# Orthogonal Model Merging

Sihan Yang[1]   Kexuan Shi[1]   Weiyang Liu[1]

## Abstract

Merging finetuned Large Language Models (LLMs) has become increasingly important for integrating diverse capabilities into a single unified model. However, prevailing model merging methods rely on linear arithmetic in Euclidean space, which often destroys the intrinsic geometric properties of pretrained weights, such as hyperspherical energy. To address this, we propose Orthogonal Model Merging (OrthoMerge), a method that performs merging operations on the Riemannian manifold formed by the orthogonal group to preserve the geometric structure of the model's weights. By mapping task-specific orthogonal matrices learned by Orthogonal Finetuning (OFT) to the Lie algebra, OrthoMerge enables a principled yet efficient integration that takes into account both the direction and intensity of adaptations. In addition to directly leveraging orthogonal matrices obtained by OFT, we further extend this approach to general models finetuned with non-OFT methods (*e.g.*, low-rank finetuning, full finetuning) via an Orthogonal-Residual Decoupling strategy. This technique extracts the orthogonal components of expert models by solving the orthogonal Procrustes problem, which are then merged on the manifold of the orthogonal group, while the remaining linear residuals are processed through standard additive merging. Extensive empirical results demonstrate the effectiveness of OrthoMerge in mitigating catastrophic forgetting and maintaining model performance across diverse tasks.

## 1. Introduction

While it has become computationally prohibitive to pretrain large foundation models from scratch (Brown et al., 2020; Chowdhery et al., 2023), recent years have witnessed an

---
[1]Department of Computer Science and Engineering, The Chinese University of Hong Kong. Correspondence to: Weiyang Liu <wyliu@cse.cuhk.edu.hk>.

*Proceedings of the 43rd International Conference on Machine Learning*, Seoul, South Korea. PMLR 306, 2026. Copyright 2026 by the author(s).

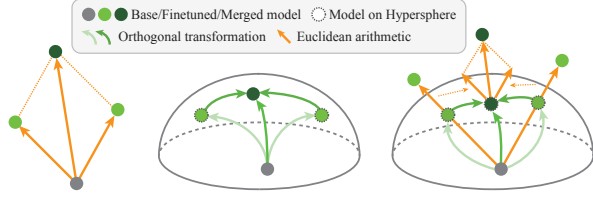

(a) Current Model Merging  (b) Orthogonal Merging  (c) Orthogonal-Residual Decoupling

*Figure 1.* A comparison among (a) current model merging, (b) orthogonal merging and (c) orthogonal-residual decoupling merging.

emerging paradigm of finetuning a large pretrained model on a downstream task using only a few demonstration examples. Despite its promising sample efficiency, the pretraining-finetuning paradigm produces many task-specific finetuned models, with the number of models scaling linearly with the number of tasks. To address this, model merging offers a scalable path to integrating domain-expert finetuned models into a single unified one without additional re-training.

Existing model merging methods (Ilharco et al., 2022; Yang et al., 2023) predominantly operate on task vectors (*i.e.*, the weight difference between the task-specific finetuned model and the base model). Early approaches aggregate task vectors using simple linear arithmetic. To mitigate interference between conflicting updates, some recent methods (Yadav et al., 2023; Kempf et al., 2023; Yu et al., 2024; Gargiulo et al., 2025) exploit vector sparsification or singular value decomposition, leveraging the low-rank structure of these updates. In all these methods, the aggregated task vector is eventually added back to the base model in Euclidean space to obtain the merged model. While these methods can be effective, they inevitably suffer from parameter conflicts across tasks. More critically, linear arithmetic in Euclidean space can destroy important geometric properties, such as hyperspherical energy (Liu et al., 2018a) (*i.e.*, the angular relationships between neurons), which are crucial for preserving model performance and representation stability.

Prior work (Qiu et al., 2023; Liu et al., 2023b; Yoshida & Miyato, 2017; Miyato et al., 2018) has shown that preserving the geometric properties and spectral norms of model weights helps mitigate catastrophic forgetting and maintain the stability of internal representations. Building on this insight, we propose a geometry-preserving model merging framework, called Orthogonal Model Merging (OrthoMerge). We posit that, compared to additive merging in Euclidean space, integrating orthogonal transformations

(*e.g.*, rotations in high-dimensional space) on a Riemannian manifold provides a more robust mechanism to merge models. For models trained with Orthogonal Finetuning (OFT) (Qiu et al., 2023), the orthogonal matrices representing these transformations are explicit. However, merging them presents a geometric dilemma: standard linear averaging destroys the orthogonality of these transformations, while the theoretically correct Riemannian geometric mean is computationally prohibitive for LLMs, requiring iterative matrix decompositions with cubic complexity $\mathcal{O}(d^3)$. To address this, we map task-specific orthogonal transformations into the Lie algebra $\mathfrak{so}(d)$, where we perform a magnitude-corrected integration that accounts for both the direction and the intensity of the adaptations. The aggregated result is then mapped back to the orthogonal group via the Cayley transform (Cayley, 1894). Furthermore, we extend this strategy to models finetuned via standard additive methods (*e.g.*, LoRA (Hu et al., 2022), full finetuning), where explicit orthogonal transformations are absent. We introduce an Orthogonal-Residual Decoupling strategy that solves the orthogonal Procrustes problem to extract the implicit orthogonal component from finetuned models. This allows us to merge the orthogonal components of the adaptation on the manifold, while handling the residuals by traditional merging in Euclidean space.

Extensive experiments across diverse domains have demonstrated that OrthoMerge can mitigate catastrophic forgetting and preserve downstream performance across tasks, consistently outperforming baselines both when merging OFT-trained models and when augmenting standard additive merging with our decoupling strategy for non-OFT-trained models. We summarize our contributions below:

- We propose OrthoMerge, a method that performs merging operations within the orthogonal group to strictly preserve the geometric structure of the model's weights.

- We introduce a magnitude-corrected averaging scheme within the Lie algebra $\mathfrak{so}(d)$ that prevents the adaptation attenuation in multi-task merging.

- We further propose an Orthogonal-Residual Decoupling framework that effectively extends OrthoMerge to models finetuned with non-OFT methods.

## 2. Related Work

**Manifold Constraints in Deep Learning**. Manifold constraints in deep learning can be systematically categorized by the stages of their application within the learning pipeline: the data representation space, the network architecture, or the optimization of weight parameters. In the realm of representation learning, constraints are primarily imposed to regularize the latent geometry, aiming to enforce specific topological structures that enhance discriminability or align with the intrinsic manifold of the data distribution (Liu et al.,

2017a; Li & He, 2025). At the architectural level, geometric constraints are integrated into the signal propagation mechanism, often by designing layers or connections that confine feature transformations to specific manifolds to preserve signal fidelity and geometric invariants (Liu et al., 2017b; 2018b; Xie et al., 2025). Finally, a diverse category of methods focuses on constraining weight matrices to Riemannian manifolds to improve optimization dynamics and generalization. This includes approaches that enforce unitary, orthogonal, or other structured constraints to ensure stability during training (Arjovsky et al., 2016; Miyato et al., 2018; Bansal et al., 2018; Liu et al., 2018a; 2021b), as well as techniques that restrict parameter updates to specific geometric groups to control semantic drift and facilitate model scaling (Liu et al., 2021a; 2023b; Qiu et al., 2023; 2025a; Xie et al., 2026; Bernstein & Newhouse, 2025).

**Model Merging**. Model merging offers a cost-effective way to build stronger models by combining expert models fine-tuned on different datasets, typically from a shared base. This modular approach supports adaptable LLM post-training and simplifies the integration of new capabilities into state-of-the-art models. There are generally two types of merging methods: those that are data-free and training-free, and those that necessitate data or training for alignment and rectification. Data-free and training-free approaches integrate finetuned models without the need for further data or retraining, making them convenient and highly efficient. These methods can be grouped into three main categories: (i) Linear interpolation methods, which merge model weights by performing arithmetic operations on task vectors (Wortsman et al., 2022; Ilharco et al., 2022; Chen et al., 2025); (ii) Sparsification-based approaches, which enhance merging performance by eliminating redundant components within task vectors (Yadav et al., 2023; Yu et al., 2024; He et al., 2024); (iii) SVD-based techniques, which leverage low-rank features identified through singular value decomposition on task vectors (Stoica et al., 2024; Gargiulo et al., 2025; Marczak et al., 2025). In contrast to the methods above, certain approaches aim to minimize the performance gap between the merged model and task-specific models through rectification strategies that require access to real data (Matena & Raffel, 2022; Jin et al., 2022), or develop some data-free proxies (Cheng et al., 2025; Shi et al., 2025) to align the merged weights optimally. All of these methods operate on task vectors and aggregate the results back to the base model, performing linear addition in Euclidean space. In contrast, our method merges models on the manifold, following a fundamentally different principle.

## 3. OrthoMerge: Orthogonal Model Merging

### 3.1. Preliminaries

Orthogonal Finetuning (Qiu et al., 2023; Liu et al., 2023b) has emerged as a powerful parameter-efficient finetuning

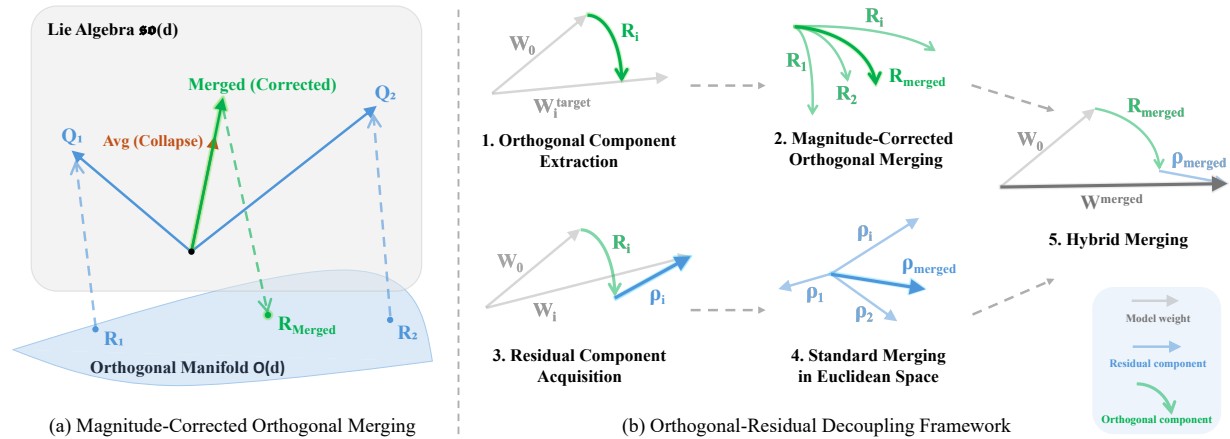

*Figure 2.* Illustration of OrthoMerge. (a) To merge orthogonal transformations, we first map them to the Lie algebra $\mathfrak{so}(d)$, perform the merging there with magnitude correction to preserve the strength of the transformations, and finally map the result back to the orthogonal group. (b) For general models, we decouple weights into orthogonal and residual components, merging them separately on the Riemannian manifold formed by the orthogonal group and in Euclidean space, respectively.

method for LLMs. While methods like LoRA introduce additive low-rank updates, OFT adapts the pretrained model by learning a multiplicative orthogonal transformation.

Given a pretrained weight matrix $\boldsymbol{W}_0 \in \mathbb{R}^{d_{in} \times d_{out}}$, OFT parameterizes the finetuned weight $\boldsymbol{W}$ as:

$$\boldsymbol{W} = \boldsymbol{R}\boldsymbol{W}_0, \quad \text{s.t. } \boldsymbol{R}^\top \boldsymbol{R} = \boldsymbol{R}\boldsymbol{R}^\top = \boldsymbol{I}, \quad (1)$$

where $\boldsymbol{R} \in \mathbb{R}^{d_{in} \times d_{in}}$ is an orthogonal matrix. To ensure strict orthogonality and differentiability during training, OFT utilizes the Cayley parameterization, constructing $\boldsymbol{R}$ from a skew-symmetric matrix $\boldsymbol{Q}$ (where $\boldsymbol{Q}^\top = -\boldsymbol{Q}$):

$$\boldsymbol{R} = (\boldsymbol{I} + \boldsymbol{Q})(\boldsymbol{I} - \boldsymbol{Q})^{-1}. \quad (2)$$

To reduce the parameter budget, $\boldsymbol{R}$ is typically structured as a block-diagonal matrix, where each block is parameterized as an orthogonal matrix independently.

### 3.2. Orthogonal Merging for OFT-trained Models

When merging multiple task-specific models finetuned via OFT, we directly obtain a set of orthogonal matrices $\{\boldsymbol{R}_i\}_{i=0}^{N-1}$ corresponding to $N$ tasks. A direct arithmetic average in the linear space, $\boldsymbol{R}_{\text{avg}} = \frac{1}{N}\sum \boldsymbol{R}_i$, destroys the orthogonality property ($\boldsymbol{R}_{\text{avg}}^\top \boldsymbol{R}_{\text{avg}} \neq \boldsymbol{I}$), thereby violating the geometric constraints which are crucial for maintaining model performance. While the geometric (Fréchet/Karcher) mean on the Riemannian manifold preserves orthogonality, it requires iterative matrix decompositions (Absil et al., 2008), leading to $\mathcal{O}(d^3)$ computational complexity per iteration for $d \times d$ matrices, which is prohibitive for LLMs with the high-dimensional hidden states. To address this, we propose to perform the merging operation within the Lie algebra $\mathfrak{so}(d)$ associated with the Lie group of orthogonal

matrices, as shown in Figure 2 (a), enabling model merging that is both efficient and orthogonality-preserving.

**Mapping to Lie Algebra $\mathfrak{so}(d)$.** We choose to map the task parameters to their skew-symmetric representations $\boldsymbol{Q}$, where orthogonal transformations can be efficiently and unconstrainedly manipulated. If explicit OFT adapter checkpoints are available, we can directly reconstruct $\boldsymbol{Q}$ from the stored parameters (typically the upper-triangular elements) by enforcing skew-symmetry ($\boldsymbol{Q}^\top = -\boldsymbol{Q}$). Alternatively, if we are only provided by the final finetuned weights $\boldsymbol{W}_{\text{ft}}$ (without the adapter weights), we first extract the potential rotation $\boldsymbol{R}$ by solving the Orthogonal Procrustes problem:

$$\boldsymbol{R} = \arg\min_{\boldsymbol{R}} \|\boldsymbol{W}_{\text{ft}} - \boldsymbol{R}\boldsymbol{W}_0\|_F, \quad \text{s.t. } \boldsymbol{R}^\top \boldsymbol{R} = \boldsymbol{I}. \quad (3)$$

The analytical solution (Gower & Dijksterhuis, 2004) can be obtained via singular value decomposition (SVD):

$$\boldsymbol{R} = \boldsymbol{U}\boldsymbol{V}^\top, \quad \text{where } \boldsymbol{U}\boldsymbol{\Sigma}\boldsymbol{V}^\top = \text{SVD}(\boldsymbol{W}_{\text{ft}}\boldsymbol{W}_0^\top). \quad (4)$$

We then obtain $\boldsymbol{Q}$ via the inverse Cayley transform:

$$\boldsymbol{Q} = (\boldsymbol{R} - \boldsymbol{I})(\boldsymbol{R} + \boldsymbol{I})^{-1}. \quad (5)$$

**Magnitude-Corrected Merging.** Geometrically, the skew-symmetric matrix $\boldsymbol{Q} \in \mathfrak{so}(d)$ acts as the infinitesimal generator of the rotation, where its Frobenius norm $\|\boldsymbol{Q}\|_F$ serves as a proxy for the angular intensity of the transformation. When merging models from diverse tasks, the task-specific updates often point in disparate directions. Consequently, a simple arithmetic mean ($\boldsymbol{Q}_{\text{mean}} = \frac{1}{N}\sum_{i=0}^{N-1} \boldsymbol{Q}_i$) can suffer from magnitude collapse due to destructive interference: conflicting components cancel out, yielding a merged update with a smaller norm. This effect can be formally character-

ized by the triangle inequality of the Frobenius norm:

$$\|\boldsymbol{Q}_{\text{mean}}\|_F = \frac{1}{N} \left\| \sum_{i=0}^{N-1} \boldsymbol{Q}_i \right\|_F \leq \frac{1}{N} \sum_{i=0}^{N-1} \|\boldsymbol{Q}_i\|_F, \quad (6)$$

with equality only when all $\boldsymbol{Q}_i$ are aligned in the same direction. In practice, misalignment across tasks makes the inequality strict, so $\|\boldsymbol{Q}_{\text{mean}}\|_F$ is typically diminished. This reduction dampens the effective adaptation strength, causing the merged model to partially lose finetuning adaptations and drift back toward the pretrained base model $\boldsymbol{W}_0$.

To mitigate this and preserve the adaptation intensity, we introduce a scaling factor $c$. We normalize the averaged adaptation by the ratio of the sum of individual norms to the norm of the sum, ensuring the merged update maintains a magnitude comparable to the individual tasks:

$$c = \frac{\sum_{i=0}^{N-1} \|\boldsymbol{Q}_i\|_F}{\left\| \sum_{i=0}^{N-1} \boldsymbol{Q}_i \right\|_F}, \quad (7)$$

where $\| \cdot \|_F$ denotes the Frobenius norm. The merged skew-symmetric matrix is then computed as:

$$\boldsymbol{Q}_{\text{merged}} = c \cdot \left( \frac{1}{N} \sum_{i=0}^{N-1} \boldsymbol{Q}_i \right). \quad (8)$$

Finally, we map $\boldsymbol{Q}_{\text{merged}}$ back to the orthogonal manifold to obtain the merged rotation $\boldsymbol{R}_{\text{merged}}$ via the Cayley transform (Eq. (2)). The final merged weight is $\boldsymbol{W}_{\text{merged}} = \boldsymbol{R}_{\text{merged}} \boldsymbol{W}_0$. This process ensures that the merged model preserves the structural benefits of orthogonal transformations while maintaining the magnitude of the adaptations.

### 3.3. Orthogonal-Residual Decoupling for Merging Non-OFT Models

For models finetuned with non-OFT methods (*e.g.*, LoRA or full finetuning), we lack explicit task-specific orthogonal matrices. The task updates exist in Euclidean space, $\boldsymbol{W}_i = \boldsymbol{W}_0 + \Delta \boldsymbol{W}_i$. To leverage the geometric stability of orthogonal merging, we propose an Orthogonal-Residual Decoupling framework, as shown in Figure 2 (b). This framework decomposes the finetuned weights into an orthogonal component (representing a structural rotation) and an additive residual component, which are merged respectively. The detailed procedure is presented in Algorithm 1.

**Orthogonal Component Extraction.** We seek an orthogonal matrix $\boldsymbol{R}_i$ that best approximates the transformation from the pretrained weights $\boldsymbol{W}_0$ to a target weight matrix $\boldsymbol{W}_i^{\text{target}}$. We consider two strategies to define this target.

*Strategy 1: Global Decoupling.* To maximize the extraction of the orthogonal component, we directly treat the finetuned

---

**Algorithm 1** Orthogonal-Residual Decoupling Merging

**Input:** Base weights $\boldsymbol{W}_0$, finetuned weights $\{\boldsymbol{W}_i\}_{i=0}^{N-1}$
**Output:** Merged weights $\boldsymbol{W}_{\text{final}}$
▷ Orthogonal component extraction.
$\{\boldsymbol{W}_i^{\text{target}}\}_{i=0}^{N-1} \leftarrow \text{GetTarget}(\boldsymbol{W}_0, \{\boldsymbol{W}_i\}_{i=0}^{N-1})$
**for** $i = 0$ **to** $N - 1$ **do**
  $\boldsymbol{U}_i, \boldsymbol{\Sigma}_i, \boldsymbol{V}_i^\top \leftarrow \text{SVD}(\boldsymbol{W}_i^{\text{target}} \boldsymbol{W}_0^\top)$
  $\boldsymbol{R}_i \leftarrow \boldsymbol{U}_i \boldsymbol{V}_i^\top$
**end for**
▷ Magnitude-corrected orthogonal merging.
**for** $i = 0$ **to** $N - 1$ **do**
  $\boldsymbol{Q}_i \leftarrow (\boldsymbol{R}_i - \boldsymbol{I})(\boldsymbol{R}_i + \boldsymbol{I})^{-1}$
**end for**
$\boldsymbol{Q}_{\text{merged}} \leftarrow \frac{1}{N} \cdot \frac{\sum_{i=0}^{N-1} \|\boldsymbol{Q}_i\|_F}{\| \sum_{i=0}^{N-1} \boldsymbol{Q}_i \|_F} \cdot \sum_{i=0}^{N-1} \boldsymbol{Q}_i$
$\boldsymbol{R}_{\text{merged}} \leftarrow (\boldsymbol{I} + \boldsymbol{Q}_{\text{merged}})(\boldsymbol{I} - \boldsymbol{Q}_{\text{merged}})^{-1}$
▷ Residual component acquisition.
**for** $i = 0$ **to** $N - 1$ **do**
  $\boldsymbol{\rho}_i \leftarrow \boldsymbol{W}_i - \boldsymbol{R}_i \boldsymbol{W}_0$
**end for**
▷ Standard merging in Euclidean space.
$\boldsymbol{\rho}_{\text{merged}} \leftarrow \text{EuclideanMerge}(\{\boldsymbol{\rho}_i\}_{i=0}^{N-1})$
▷ Hybrid merging.
$\boldsymbol{W}_{\text{final}} \leftarrow \boldsymbol{R}_{\text{merged}} \boldsymbol{W}_0 + \boldsymbol{\rho}_{\text{merged}}$

---

weight itself as the target:

$$\boldsymbol{W}_i^{\text{target}} = \boldsymbol{W}_i. \quad (9)$$

*Strategy 2: Conflict-Aware Decoupling.* This strategy isolates the components of the update that conflict with the consensus direction at the neuron level. Let $\boldsymbol{\tau}_i = \boldsymbol{W}_i - \boldsymbol{W}_0$ denote the task vector. We compute the average task vector $\boldsymbol{\tau}_{\text{mean}} = \frac{1}{N} \sum \boldsymbol{\tau}_i$ and identify conflicting neurons (columns) for task $i$ where the local update opposes the global trend:

$$\mathcal{S}_i = \{j \mid \cos(\boldsymbol{\tau}_i[:, j], \boldsymbol{\tau}_{\text{mean}}[:, j]) < 0\}. \quad (10)$$

We construct the conflict-specific target by adding only these neuron-level conflicting updates (where columns not in $\mathcal{S}_i$ are zeroed out in $\boldsymbol{\tau}_i^{\text{conf}}$) to the base weights:

$$\boldsymbol{W}_i^{\text{target}} = \boldsymbol{W}_0 + \boldsymbol{\tau}_i^{\text{conf}}. \quad (11)$$

*Orthogonal Approximation via Procrustes Analysis.* Given the target matrix $\boldsymbol{W}_i^{\text{target}}$ defined by one of the strategies above, we solve for the orthogonal matrix $\boldsymbol{R}_i$ via the following Orthogonal Procrustes problem:

$$\min_{\boldsymbol{R}_i} \|\boldsymbol{W}_i^{\text{target}} - \boldsymbol{R}_i \boldsymbol{W}_0\|_F, \quad \text{s.t.} \quad \boldsymbol{R}_i^\top \boldsymbol{R}_i = \boldsymbol{I}. \quad (12)$$

The closed-form solution is given by $\boldsymbol{R}_i = \boldsymbol{U} \boldsymbol{V}^\top$, where $\boldsymbol{U} \boldsymbol{\Sigma} \boldsymbol{V}^\top = \text{SVD}(\boldsymbol{W}_i^{\text{target}} \boldsymbol{W}_0^\top)$.

**Magnitude-Corrected Orthogonal Merging for the Orthogonal Component.** Once the orthogonal matrices

$\{\boldsymbol{R}_i\}_{i=0}^{N-1}$ are extracted, we convert each $\boldsymbol{R}_i$ to its Lie algebra representation $\boldsymbol{Q}_i$ via the inverse Cayley transform and obtain the merged orthogonal transformation $\boldsymbol{R}_{\text{merged}}$ using the *Magnitude-Corrected Merging* described in Section 3.2.

**Residual Component Acquisition.** Simultaneously, we calculate the additive residuals for each task, which capture the information not modeled by the rotation:

$$\boldsymbol{\rho}_i = \boldsymbol{W}_i - \boldsymbol{R}_i \boldsymbol{W}_0. \tag{13}$$

**Standard Merging in Euclidean Space for the Residual Component.** The residuals are merged using conventional model merging methods that operate in Euclidean space (*e.g.*, Task arithmetic or TIES-Merging) to obtain $\boldsymbol{\rho}_{\text{merged}}$.

**Hybrid Merging.** The final merged model reconstructs the weight by applying the merged orthogonal transformation to the base and adding the merged residual:

$$\boldsymbol{W}_{\text{final}} = \boldsymbol{R}_{\text{merged}} \boldsymbol{W}_0 + \boldsymbol{\rho}_{\text{merged}}. \tag{14}$$

By decoupling the weight updates into rotational and additive components, this approach allows us to merge structural transformations on the orthogonal manifold for geometric stability, while retaining the plasticity of additive merging for scaling and residual adjustments.

## 4. Intriguing Insights and Discussions

**Better Loss Landscape with OrthoMerge**. In Figure 3, we illustrate the locations of the base model, as well as models merged by the classical Euclidean method TA and our OrthoMerge, on the joint loss landscape of all merged tasks. As shown in Figure 3, OrthoMerge achieves a more favorable loss location in terms of both optimization direction and magnitude compared to TA. The results validate that performing merging on the Riemannian manifold induced by the orthogonal group can better preserve model knowledge and reduce destructive interference.

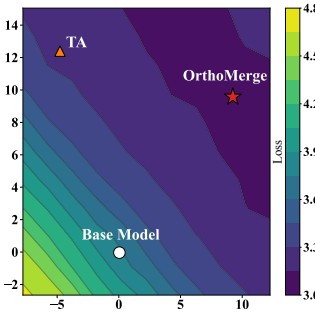

*Figure 3.* Loss landscape of the base model, TA and OrthoMerge.

**Hyperspherical Energy Preservation**. Prior studies (Qiu et al., 2023; 2025b; Liu et al., 2023b) have shown that preserving the hyperspherical energy plays a critical role in mitigating catastrophic forgetting. OrthoMerge naturally inherits this desirable property. Specifically, our method merges task-specific adaptations into one orthogonal matrix $\boldsymbol{R}_{\text{merged}}$. Since orthogonal transformations preserve inner products and vector norms, applying $\boldsymbol{R}_{\text{merged}}$ to the base model does not alter the hyperspherical energy. In contrast,

Euclidean-space merging methods ($\sum_i (\boldsymbol{W}_i - \boldsymbol{W}_0)$) introduce unconstrained linear perturbations to the weight space, which distorts the hyperspherical geometry. In Table 1, 2, 3, OrthoMerge not only achieves the best in-domain performance, but also consistently outperforms Euclidean-space merging methods on out-of-distribution tasks, indicating reduced catastrophic forgetting.

**Spectral Regularization**. Another special property of OrthoMerge is its implicit regularization on the spectral norm (Yoshida & Miyato, 2017; Miyato et al., 2018). As described above, the proposed method OrthoMerge performs merging in the Lie algebra by aggregating task-specific skew-symmetric matrices $\{\boldsymbol{Q}_i\}$, and then maps the merged result back to the orthogonal manifold via the Cayley transformation (Eq. 8). Since the resulting transformation $\boldsymbol{R}_{\text{merged}}$ is guaranteed to be orthogonal, the merged weights $\boldsymbol{W}_{\text{merged}} = \boldsymbol{R}_{\text{merged}} \boldsymbol{W}_0$ preserve the spectral norm of the base model, i.e., $\|\boldsymbol{W}_{\text{merged}}\|_2 = \|\boldsymbol{W}_0\|_2$, regardless of the number of merged tasks $N$ or the diversity of task directions. In contrast to conventional Euclidean-space merging methods, which may amplify or attenuate the spectral norm as tasks accumulate, OrthoMerge enforces a geometric constraint that alleviates spectral drift.

## 5. Experiments and Results

We evaluate OrthoMerge across two distinct scenarios: (1) merging models finetuned with Orthogonal Finetuning (OFT), where we directly apply our Orthogonal Merging operations; and (2) merging models finetuned with non-orthogonal methods (*e.g.*, LoRA, full finetuning) via our proposed Orthogonal-Residual Decoupling strategy.

### 5.1. Merging OFT-Finetuned Models

#### 5.1.1. EXPERIMENTAL SETUP

**Models for Merging.** To rigorously evaluate the effectiveness of orthogonal merging, we finetuned the base models Llama-3.1-8B (Grattafiori et al., 2024) and Qwen2.5-3B (Qwen et al., 2025) using both LoRA and OFT to obtain task-specific models, which were subsequently merged. The models were finetuned on five datasets: ScienceQA (Lu et al., 2022), CommonsenseQA (Talmor et al., 2019), Social-IQA (Sap et al., 2019), Magicoder-OSS-Instruct (Wei et al., 2023), and NuminaMath-TIR (Li et al., 2024), covering science, commonsense, social knowledge, coding, and math.

**Evaluation.** We assess the performance of the merged models on the corresponding test sets of ScienceQA, CommonsenseQA, and Social-IQA, as well as MATH500 (Hendrycks et al., 2021) for math and HumanEval+ (Liu et al., 2023a) for coding. Furthermore, to assess whether the merging method preserves the base model's original capabilities, we use M-ARC (Pomerenke et al., 2025) to evaluate multilin-

| | Model | MATH500 | HUMANEVAL+ | ScienceQA | CommonsenseQA | Social-IQA | Avg. | M-ARC | AGIEval | Avg. |
|---|---|---|---|---|---|---|---|---|---|---|
| | Llama-3.1-8B | 17.80 | 21.52 | 6.12 | 70.60 | 48.11 | 32.83 | 40.85 | 35.76 | 38.31 |
| **LoRA** | Task-specific FT | 19.40 | 38.48 | 26.44 | **83.78** | **58.39** | **45.30** | – | – | – |
| | TA | 21.00 | 34.20 | 29.90 | 80.18 | 54.15 | 43.89 | 43.80 | 37.90 | **40.85** |
| | TIES | 18.80 | **38.65** | 20.68 | 82.15 | 56.96 | 43.45 | **44.57** | 37.09 | 40.83 |
| | DARE | **22.60** | 34.26 | 29.68 | 80.10 | 53.84 | 44.10 | 43.80 | **37.86** | 40.83 |
| | TSV-M | 0.60 | 31.09 | **30.62** | 83.05 | 58.03 | 40.68 | 43.44 | 34.51 | 38.98 |
| **OFT** | Task-specific FT | 19.00 | 38.54 | 26.75 | **82.47** | **56.81** | 44.71 | – | – | – |
| | TA | 22.40 | 33.29 | 22.62 | 76.82 | 52.20 | 41.46 | 43.03 | 38.53 | 40.78 |
| | TIES | 21.60 | 39.93 | 29.41 | 78.38 | 53.74 | 44.61 | 43.01 | 37.85 | 40.43 |
| | DARE | 20.80 | 31.21 | 22.48 | 77.81 | 52.10 | 40.88 | 43.08 | 38.47 | 40.78 |
| | TSV-M | 18.40 | 37.86 | **33.72** | 79.85 | 55.02 | 44.97 | 44.00 | 36.96 | 40.48 |
| | OrthoMerge (ours) | **24.60** | **38.90** | 32.01 | 80.59 | 55.17 | **46.25** | **44.75** | **38.84** | **41.80** |

*Table 1.* Performance of different merging methods on task-specific models finetuned from Llama-3.1-8B using LoRA and OFT, evaluated on task-specific benchmarks, out-of-domain, and general evaluation datasets.

gual ability (which is not covered by any of the merged task-specific models) and AGIEval (Zhong et al., 2024) to assess overall performance in real-world scenarios.

**Baselines.** We compare our proposed method against baselines applied to both LoRA and OFT models, including TA (Ilharco et al., 2022), TIES-Merging (Yadav et al., 2023), DARE (Yu et al., 2024), and TSV-M (Gargiulo et al., 2025). Note that for OFT models, all baseline methods perform merging directly in the weight space ($W$).

### 5.1.2. RESULTS AND DISCUSSION

The results for Llama-3.1-8B are presented in Table 1, while the results for Qwen2.5-3B are provided in the Appendix. Overall, OrthoMerge consistently outperforms all Euclidean-space baselines for both LoRA and OFT models on both in-domain merged tasks and out-of-domain generalization benchmarks. On task-specific benchmarks, OrthoMerge achieves the highest average accuracy of 46.25%, notably surpassing the average performance of the individual experts (44.71%). This suggests constructive synergy, rather than destructive interference, during the model merging process. Importantly, this advantage extends to out-of-domain generalization: OrthoMerge attains the highest average accuracy of 41.80% on M-ARC and AGIEval, indicating that preserving the intrinsic geometric structure of weights via the manifold constraint provides a more robust mechanism for retaining general capabilities while integrating diverse task-specific skills.

### 5.2. Merging Non-OFT Models

#### 5.2.1. EXPERIMENTAL SETUP

**Models for Merging.** To demonstrate the generalizability of our proposed framework, we apply our Orthogonal-Residual Decoupling strategy to models finetuned with non-OFT methods. Specifically, we merge five task-specific models, which are provided by MergeBench (He et al., 2025), each trained on a distinct domain, using Llama-3.2-3B (Grattafiori et al., 2024) as the base model.

**Evaluation.** The evaluation spans five areas (following MergeBench): instruction following (IFEval (Zhou et al., 2023)), math (GSM8k (Cobbe et al., 2021) with CoT), coding (HumanEval+ and MBPP+ (Austin et al., 2021)), multilingual ability (aggregated ARC, *i.e.* in multilingual settings (Pomerenke et al., 2025)), and safety (aggregated WildGuardTest (Han et al., 2024), HarmBench (Mazeika et al., 2024), XSTest (Röttger et al., 2024), and DoAnything-Now (Shen et al., 2024) scores). Additionally, to assess whether the merging method preserves the base model's original capabilities, we use MMLU (Hendrycks et al., 2020) (excluding math- and coding-related subsets) to evaluate performance on tasks in STEM, social sciences, humanities, etc., none of which are covered by the task-specific models used for merging. We also use AGIEval to assess overall performance in real-world scenarios.

**Baselines.** For comparison, we integrate our proposed Orthogonal-Residual Decoupling framework (with the global target denoted as +OrthoMerge-G and the conflict-aware target as +OrthoMerge-C) into three types of standard merging baselines: TA (linear interpolation), TIES (sparsification-based), and TSV-M (SVD-based).

### 5.2.2. RESULTS AND DISCUSSION

The results for merging task-specific models finetuned from Llama-3.2-3B are summarized in Table 2. Our Orthogonal-Residual Decoupling framework enhances the performance of standard merging methods (TA, TIES, and TSV-M) across task-specific, out-of-domain, and general benchmarks. Specifically, integrating the Conflict-Aware strategy (+OrthoMerge-C) with TSV-M achieves the highest average task performance of 42.07%. Notably, +OrthoMerge-G improves TA by 1.32%, and +OrthoMerge-C significantly boosts TIES by 2.41%. These results indicate that decoupling orthogonal structural updates from additive task-specific updates enables more effective integration of task-specific capabilities. Moreover, while standard merging methods often involve a trade-off between task specialization and generalization, our approach alleviates this issue.

| Model | Instruction | Math | Coding | Multilingual | Safety | Avg. | MMLU† | AGIEval | Avg. |
|---|---|---|---|---|---|---|---|---|---|
| Llama-3.2-3B | 7.58 | 28.51 | 27.44 | 40.72 | 31.41 | 27.18 | 55.00 | 31.05 | 43.03 |
| Task-specific FT | 39.56 | 69.83 | 44.33 | 41.73 | 80.46 | 55.18 | – | – | – |
| TA | 10.53 | 40.40 | 37.22 | 42.26 | 40.40 | 34.16 | 55.35 | 32.67 | 44.01 |
| +OrthoMerge-C | 9.80 | 40.03 | 37.75 | **42.29** | 41.69 | 34.31 (+0.15) | **55.76** | 32.69 | **44.23** (+0.22) |
| +OrthoMerge-G | **11.09** | **43.90** | **37.96** | 42.06 | **42.41** | **35.48** (+1.32) | 55.57 | **32.85** | 44.21 (+0.20) |
| TIES | 20.89 | 50.80 | 39.61 | 42.11 | 42.38 | 39.16 | 54.52 | 33.88 | 44.20 |
| +OrthoMerge-C | **25.32** | **55.80** | 40.21 | **42.26** | **44.24** | **41.57** (+2.41) | **55.36** | **34.01** | **44.69** (+0.49) |
| +OrthoMerge-G | 20.33 | 51.48 | **40.28** | 42.08 | 42.51 | 39.34 (+0.18) | 54.89 | 33.45 | 44.17 (-0.03) |
| TSV-M | 19.40 | 55.72 | 40.44 | 42.20 | 49.82 | 41.52 | 55.49 | 33.56 | 44.53 |
| +OrthoMerge-C | **19.59** | **55.88** | **41.69** | **42.32** | **50.87** | **42.07** (+0.55) | 55.57 | **33.79** | 44.68 (+0.15) |
| +OrthoMerge-G | 18.67 | 51.71 | 40.65 | 41.89 | 47.23 | 40.03 (-1.49) | **55.92** | 33.56 | **44.74** (+0.21) |

*Table 2.* Performance of merging task-specific models (provided by MergeBench) finetuned from Llama-3.2-3B. Results compare standard merging baselines and their combinations with our OrthoMerge variants (+OrthoMerge-C, +OrthoMerge-G). "MMLU†" denotes MMLU evaluated with math- and coding-related subsets excluded.

| Model | MMSI-Bench | EmbSpatial | MMMU$_{\text{Med}}$ | PathVQA | OCRBench | CharXiv | Avg. | IFEval | MMBench | Avg. |
|---|---|---|---|---|---|---|---|---|---|---|
| Qwen2.5-VL-7B-Instruct | 27.80 | 69.97 | 53.10 | 66.30 | 84.70 | 67.20 | 61.51 | 61.74 | 84.02 | 72.88 |
| Task-specific FT | 32.60 | 70.58 | 55.17 | 66.81 | 85.00 | 72.50 | 63.78 | – | – | – |
| TA | 28.70 | 71.25 | 52.41 | 67.76 | 83.70 | 68.40 | 62.04 | 60.12 | 83.84 | 71.98 |
| +OrthoMerge-C | 29.00 | 71.26 | 54.48 | 67.94 | **84.60** | **69.10** | 62.73 (+0.69) | **61.00** | 83.85 | **72.43** (+0.45) |
| +OrthoMerge-G | **29.90** | **71.29** | **57.24** | **68.26** | 83.50 | 68.70 | **63.15** (+1.11) | 60.81 | **83.94** | 72.38 (+0.40) |
| TIES | 32.40 | 71.57 | 57.93 | 67.61 | 82.60 | 69.10 | 63.54 | 55.15 | 83.17 | 69.16 |
| +OrthoMerge-C | **32.50** | 71.59 | **58.62** | **68.50** | 83.10 | **69.90** | **64.04** (+0.50) | 54.90 | 83.59 | 69.25 (+0.09) |
| +OrthoMerge-G | 32.40 | **71.76** | 57.93 | 67.67 | **84.30** | 69.70 | 63.96 (+0.42) | **56.19** | **84.19** | **70.19** (+1.03) |
| TSV-M | 31.40 | 71.95 | 54.86 | 68.20 | 83.20 | 69.00 | 63.10 | 54.90 | 83.07 | 68.99 |
| +OrthoMerge-C | 31.10 | 72.31 | 54.80 | **68.74** | 83.30 | **69.90** | 63.36 (+0.26) | 54.90 | 83.42 | 69.16 (+0.17) |
| +OrthoMerge-G | **32.30** | **72.53** | **55.86** | 68.29 | **83.70** | 69.10 | **63.63** (+0.53) | **57.86** | **83.85** | **70.86** (+1.87) |

*Table 3.* Performance of merging task-specific vision-language models finetuned from Qwen2.5-VL-7B-Instruct. Results compare standard merging baselines as well as their combinations with our OrthoMerge variants (+OrthoMerge-C, +OrthoMerge-G).

For example, whereas TIES slightly reduces the MMLU† score compared to the base model (54.52% vs. 55.00%), TIES+OrthoMerge-C recovers and even exceeds this performance (55.36%), demonstrating that merging models on the manifold of the orthogonal group helps preserve the knowledge of the pre-trained model.

### 5.2.3. VISION-LANGUAGE MODEL EXTENSION

**Setup.** To further validate the effectiveness of the Orthogonal-Residual Decoupling framework, we merge three models finetuned from Qwen2.5-VL-7B-Instruct (Bai et al., 2025): SenseNova-SI-1.1-Qwen2.5-VL-7B (Cai et al., 2025) (spatial reasoning), olmOCR-2-7B-1025 (Poznanski et al., 2025) (optical character recognition), and HuatuoGPT-Vision-7B (Chen et al., 2024) (medical multimodal). For evaluation, we assess spatial reasoning with MMSI-Bench (Yang et al., 2025) and EmbSpatial (Du et al., 2024), multimodal medical QA with MMMU's medical subset and PathVQA (He et al., 2020), and OCR performance with OCRBench (Liu et al., 2024b) and CharXiv (Wang et al., 2024). Additionally, we test whether it retains the instruction following ability of the original models using IFEval and evaluate the general capabilities of the merged model on MMBench (Liu et al., 2024a).

**Results.** In Table 3, we show the model merging results for the vision-language domain, verifying that the geometric principles of OrthoMerge are modality-agnostic. Across all baselines, augmenting the merging process with our decoupling strategies results in superior multi-task performance. TIES+OrthoMerge-C achieves the highest average accuracy on task-specific benchmarks (64.04%), surpassing the individual task-specific models' average (63.78%) and the vanilla TIES baseline (63.54%). This suggests that our method effectively synthesizes diverse capabilities into a single unified model. Crucially, our method demonstrates superior stability in preserving the model's general instruction following and multimodal capabilities. While vanilla TIES suffers a significant drop in IFEval scores (from 61.74% to 55.15%), TIES+OrthoMerge-G retains a much higher performance (56.19%), and TSV-M+OrthoMerge-G achieves 57.86%, surpassing TSV-M by 2.96%. Similarly, on the general MMBench evaluation, our methods consistently outperform their respective baselines. This indicates that by respecting the geometry of the weights during merging, OrthoMerge minimizes the destructive interference that typically degrades generalist abilities in Multimodal LLMs.

### 5.3. Ablation Study and Analysis

**Effect of Orthogonal Transformation Merging Strategies.** We investigate the impact of different orthogonal transformation merging strategies on model performance, as summarized in Table 4. Directly averaging orthogonal

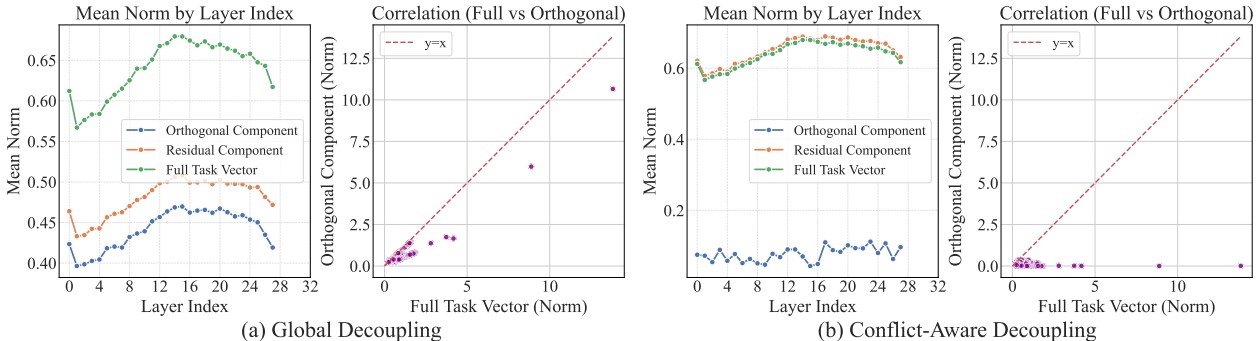

(a) Global Decoupling  (b) Conflict-Aware Decoupling

*Figure 4.* Comparison of norm distributions between different decoupling strategies applied to the models from Section 5.2.1. (a) Norm statistics using the Global Decoupling strategy. (b) Norm statistics using the Conflict-Aware Decoupling strategy.

| Model | Math | Coding | Science | Commonsense | Social | Avg. |
|---|---|---|---|---|---|---|
| Llama-3.1-8B | 17.80 | 21.52 | 6.12 | 70.60 | 48.11 | 32.83 |
| Task-specific FT | 19.00 | 38.54 | 26.75 | 82.47 | 56.81 | 44.71 |
| Simple Average R | 21.40 | 30.06 | 18.88 | 76.82 | 50.51 | 39.53 |
| Sequential Product R | 8.60 | 33.05 | 23.65 | **80.84** | **57.68** | 40.76 |
| Simple Average Q | 22.00 | 30.85 | 23.11 | 76.99 | 52.10 | 41.01 |
| OrthoMerge | **24.60** | **38.90** | **32.01** | 80.59 | 55.17 | **46.25** |

*Table 4.* Comparison of different orthogonal transformation merging strategies across task-specific benchmarks (the same benchmark settings as in Section 5.1).

matrices in Euclidean space (Simple Average R) yields suboptimal results (39.53%), as this operation destroys the orthogonality property, thereby violating the intrinsic geometric constraints of the weights. While sequential multiplication (Sequential Product R) maintains orthogonality, it suffers from severe instability, evidenced by the collapse in MATH500 performance to 8.60%, due to the excessive rotation. Performing the average within the Lie algebra (Simple Average Q) ensures the merged result remains a valid rotation and improves stability (41.01%), yet it still significantly underperforms the individual experts. This is attributed to magnitude collapse, where conflicting update directions cancel out, dampening the overall strength of the adaptation. Finally, OrthoMerge significantly outperforms all variants (46.25%), confirming that our magnitude-corrected averaging scheme is essential for preserving both the valid geometric structure and the intensity of task-specific adaptations.

**Performance across Different Numbers of Tasks.** As shown in Figure 5, OrthoMerge consistently outperforms all other methods when merging different numbers of task-specific finetuned models, demonstrating its effectiveness and robustness compared to alternative merging strategies. We also observe that the performance gain becomes increasingly more signifi-

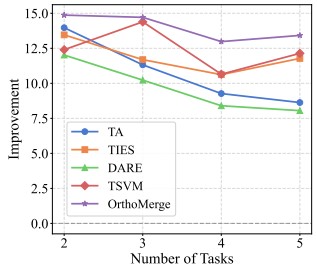

*Figure 5.* Performance gain vs. number of tasks of different merging methods over base model.

cant when the number of merged tasks gets larger. Detailed experimental results are provided in the Appendix B.3.

**Analysis of the Norm Distribution of Orthogonal Components.** To validate the effectiveness of our decoupling strategies, we compare the norm distributions of the orthogonal and residual components, as well as the original full task vector, under different decoupling approaches. Here, the orthogonal component refers to the difference between the orthogonally transformed weights and the original weights, i.e., $R_i W_0 - W_0$. As illustrated in Figure 4 (a), with the Global Decoupling strategy, the norm of the orthogonal component is positively correlated with that of the full task vector. This indicates that the strategy successfully extracts a substantial portion of parameter updates as rotational transformations. Conversely, as shown in Figure 4 (b), the orthogonal component extracted by Conflict-Aware Decoupling exhibits a minimal norm. This implies that the vast majority of update information is preserved within the residual component, with the orthogonal transformation serving only to finetune the structure in conflicting parameter updates.

## 6. Concluding Remarks

This work introduces a novel model merging paradigm that operates the model weights within the orthogonal group. Specifically, we propose OrthoMerge, a geometrically principled framework that shifts the integration of expert models from Euclidean space to the Riemannian manifold of the orthogonal group. By leveraging magnitude-corrected merging in the Lie algebra and a novel Orthogonal-Residual Decoupling strategy, OrthoMerge enables robust and generalizable merging of diverse task-specific finetuned models. Extensive experiments across language and multimodal domains demonstrate that OrthoMerge consistently surpasses traditional Euclidean-space merging methods, sustaining strong performance across varied tasks and effectively mitigating catastrophic forgetting. OrthoMerge lays a solid foundation for stable, generalizable, and scalable composition of comprehensive intelligence from specialized models. comprehensive intelligence from specialized models.

## Impact Statement

This paper presents work whose goal is to advance the field of Machine Learning. There are many potential societal consequences of our work, none which we feel must be specifically highlighted here.

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

# Appendix

**Table of Contents**

# A. Additional Implementation Details

## A.1. Training Hyperparameters

We summarize the training hyperparameters for both LoRA and OFT methods in Tables 5 and 6, respectively. For hyperparameters shared by both methods, we use identical values, while for method-specific hyperparameters, we adopt commonly used settings.

| Hyperparameter | Value |
| --- | --- |
| Rank ($r$) | 32 |
| $\alpha$ | 64 |
| Dropout | 0.05 |
| Batch size | 64 |
| Learning rate | $2 \times 10^{-4}$ |
| Epochs | 2 |

*Table 5.* LoRA Training Hyperparameters

| Hyperparameter | Value |
| --- | --- |
| Block size | 32 |
| Dropout | 0.05 |
| Batch size | 64 |
| Learning rate | $2 \times 10^{-4}$ |
| Epochs | 2 |

*Table 6.* OFT Training Hyperparameters

## A.2. Detailed Procedure of Orthogonal-Residual Decoupling

We summarize the detailed procedure of Orthogonal-Residual Decoupling in Algorithm 2. This is the full version of Algorithm 1 from the main text, and it provides the complete process for orthogonal component extraction.

## A.3. Loss Landscape Visualization Method

To plot the loss landscape in Figure 3, we first flatten the parameters of the base model and the merged models into vectors. We then construct a two-dimensional plane in parameter space by taking the directions from the base model to each merged model and orthonormalizing them via Gram-Schmidt. For each point on a regular 2D grid within this plane, we interpolate the model parameters accordingly and evaluate the loss on the training dataset. The resulting loss values are visualized as a contour plot, with the base and merged models projected onto the same plane for direct comparison. This approach provides an intuitive view of how different merging methods influence the local optimization landscape.

## A.4. Task-Specific Models Provided by MergeBench

MergeBench provides five task-specific models, each finetuned on a distinct dataset covering a representative domain. The details of the datasets used for finetuning are as follows:

- **Instruction-following:** The model is finetuned on the TULU-3 persona IF dataset (Lambert et al., 2024), which focuses on precise instruction following.

- **Mathematics:** Two datasets are used for the mathematics domain. DART-Math (Tong et al., 2024) contains difficulty-aware math problems, while NuminaMath-TIR (Li et al., 2024) consists of math competition problems.

- **Multilingual:** The multilingual model is trained on the Aya dataset (Singh et al., 2024), which is human-curated and covers data in 65 languages.

- **Coding:** The coding model is finetuned on Magicoder (Wei et al., 2023), a dataset of coding problems generated from real-world code snippets.

- **Safety:** For the safety domain, two datasets are used: WildGuardMix (Han et al., 2024), which contains both vanilla and adversarial safety prompts, and WildJailbreak (Jiang et al., 2024), a dataset with both harmful and benign prompts.

These diverse datasets ensure that the task-specific models from MergeBench comprehensively cover important domains for model merging experiments.

---

**Algorithm 2** Orthogonal-Residual Decoupling Merging (Full Version)

---

**Input:** Base weights $\boldsymbol{W}_0$, finetuned weights $\{\boldsymbol{W}_i\}_{i=0}^{N-1}$, strategy flag strategy $\in$ {use_conflict_aware, use_global}

**Output:** Merged weights $\boldsymbol{W}_{\text{final}}$

$\triangleright$ Orthogonal component extraction.

**for** $i = 0$ **to** $N - 1$ **do**
  $\boldsymbol{\tau}_i \leftarrow \boldsymbol{W}_i - \boldsymbol{W}_0$
**end for**

**if** strategy = use_conflict_aware **then**
  $\boldsymbol{\tau}_{\text{mean}} \leftarrow \frac{1}{N} \sum_{i=0}^{N-1} \boldsymbol{\tau}_i$
  **for** $i = 0$ **to** $N - 1$ **do**
    Initialize $\boldsymbol{\tau}_i^{\text{conf}}$ as zero matrix with same shape as $\boldsymbol{W}_0$
    **for** each neuron/column index $j$ **do**
      $c_{i,j} \leftarrow \cos(\boldsymbol{\tau}_i[:,j], \boldsymbol{\tau}_{\text{mean}}[:,j])$
      **if** $c_{i,j} < 0$ **then**
        $\boldsymbol{\tau}_i^{\text{conf}}[:,j] \leftarrow \boldsymbol{\tau}_i[:,j]$
      **end if**
    **end for**
    $\boldsymbol{W}_i^{\text{target}} \leftarrow \boldsymbol{W}_0 + \boldsymbol{\tau}_i^{\text{conf}}$
  **end for**
**else if** strategy = use_global **then**
  **for** $i = 0$ **to** $N - 1$ **do**
    $\boldsymbol{W}_i^{\text{target}} \leftarrow \boldsymbol{W}_i$
  **end for**
**end if**

**for** $i = 0$ **to** $N - 1$ **do**
  $\boldsymbol{U}_i, \boldsymbol{\Sigma}_i, \boldsymbol{V}_i^\top \leftarrow \text{SVD}(\boldsymbol{W}_i^{\text{target}} \boldsymbol{W}_0^\top)$
  $\boldsymbol{R}_i \leftarrow \boldsymbol{U}_i \boldsymbol{V}_i^\top$
**end for**

$\triangleright$ Magnitude-corrected orthogonal merging.

**for** $i = 0$ **to** $N - 1$ **do**
  $\boldsymbol{Q}_i \leftarrow (\boldsymbol{R}_i - \boldsymbol{I})(\boldsymbol{R}_i + \boldsymbol{I})^{-1}$
**end for**

$\boldsymbol{Q}_{\text{merged}} \leftarrow \frac{1}{N} \cdot \frac{\sum_{i=0}^{N-1} \|\boldsymbol{Q}_i\|_F}{\left\|\sum_{i=0}^{N-1} \boldsymbol{Q}_i\right\|_F} \cdot \sum_{i=0}^{N-1} \boldsymbol{Q}_i$

$\boldsymbol{R}_{\text{merged}} \leftarrow (\boldsymbol{I} + \boldsymbol{Q}_{\text{merged}})(\boldsymbol{I} - \boldsymbol{Q}_{\text{merged}})^{-1}$

$\triangleright$ Residual component acquisition.

**for** $i = 0$ **to** $N - 1$ **do**
  $\boldsymbol{\rho}_i \leftarrow \boldsymbol{W}_i - \boldsymbol{R}_i \boldsymbol{W}_0$
**end for**

$\triangleright$ Standard merging in Euclidean space.

$\boldsymbol{\rho}_{\text{merged}} \leftarrow \text{EuclideanMerge}\left(\{\boldsymbol{\rho}_i\}_{i=0}^{N-1}\right)$

$\triangleright$ Hybrid merging.

$\boldsymbol{W}_{\text{final}} \leftarrow \boldsymbol{R}_{\text{merged}} \boldsymbol{W}_0 + \boldsymbol{\rho}_{\text{merged}}$

---

# B. Additional Experimental Results

## B.1. Efficiency Analysis

We analyze the computational overhead of solving the orthogonal Procrustes problem. For Llama-3.2-3B, the SVD-based Procrustes solver takes 19 minutes and 22 seconds per task on a single GPU. Since model merging is a one-time offline procedure and does not introduce any additional cost during deployment, this overhead is generally acceptable in practice. To further improve efficiency, we implement a faster variant based on the Newton-Schulz iteration, following Muon (Jordan et al., 2024). This reduces the runtime to 1 minute and 14 seconds, yielding a substantial speedup over the SVD-based implementation.

Interestingly, the Newton-Schulz variant achieves comparable or even better performance than the SVD-based Procrustes solver, as shown in Table 7. We note that the SVD solver used in practice is itself an iterative numerical procedure rather than an exact analytical solution; for example, PyTorch computes SVD approximately using Jacobi-type iterations. Moreover, since pretrained model weights are typically stored in low precision, e.g., `float16`, additional numerical precision in the orthogonal factor does not necessarily translate into better downstream performance.

Therefore, strict orthogonalization is not always required for strong fusion performance. The Newton-Schulz approximation may provide a more conservative orthogonal component, reducing over-rotation and improving the balance between structural alignment and task-specific adaptation. In addition, its approximation error may act as an implicit regularizer, preventing task-specific noise from being overly absorbed into the orthogonal branch. Similar phenomena have also been observed in recent orthogonalization-based optimizers, where more accurate orthogonalization does not always lead to better generalization (Jordan et al., 2024).

| Model | Inst. | Math | Code | Multi. | Safety | Avg. |
|---|---|---|---|---|---|---|
| TA | 10.53 | 40.40 | 37.22 | 42.26 | 40.40 | 34.16 |
| + OrthoMerge-C (SVD) | 9.80 | 40.03 | 37.75 | 42.29 | 41.69 | 34.31 |
| + OrthoMerge-C (Newton-Schulz) | 10.72 | 39.88 | 37.63 | 42.23 | 41.12 | 34.32 |
| + OrthoMerge-G (SVD) | 11.09 | 43.90 | 37.96 | 42.06 | 42.41 | 35.48 |
| + OrthoMerge-G (Newton-Schulz) | 12.08 | 44.55 | 38.05 | 42.41 | 43.07 | 36.03 |

*Table 7.* Efficiency and performance comparison between the SVD-based Procrustes solver and the Newton-Schulz variant on Llama-3.2-3B. The Newton-Schulz variant reduces runtime from 19m22s to 1m14s per task.

## B.2. Full Results on Qwen2.5-3B for Merging OFT-Finetuned Models

Table 8 reports the detailed performance of all evaluated merging methods on the Qwen2.5-3B base model across five representative benchmarks: MATH500, HUMANEVAL+, ScienceQA, CommonsenseQA, and Social-IQA. We provide results for both LoRA- and OFT-based task-specific models, as well as all baselines and our proposed OrthoMerge method.

Consistent with the results on Llama-3.1-8B, OrthoMerge achieves strong and robust performance when merging Qwen2.5-3B models. Notably, OrthoMerge yields the highest average score among all methods. These results further validate the effectiveness and generalizability of our orthogonal merging strategy across different base models.

| | Model | MATH500 | HUMANEVAL+ | ScienceQA | CommonsenseQA | Social-IQA | Avg. |
|---|---|---|---|---|---|---|---|
| | Qwen2.5-3B | 26.80 | 6.50 | 73.97 | 76.99 | 49.95 | 46.84 |
| **LoRA** | Task-specific FT | 27.80 | 14.21 | 92.04 | 82.96 | 52.41 | 57.83 |
| | TA | 30.80 | 12.20 | 83.86 | 80.75 | 53.43 | 52.21 |
| | TIES | 31.60 | 21.03 | 83.59 | 81.24 | 53.89 | 54.27 |
| | DARE | 33.40 | 10.55 | 84.35 | 80.84 | 53.38 | 52.50 |
| | TSV-M | 7.80 | 36.09 | 90.47 | 79.61 | 51.94 | 53.18 |
| **OFT** | Task-specific FT | 27.80 | 14.75 | 92.99 | 83.46 | 52.87 | 58.46 |
| | TA | 27.60 | 10.60 | 84.94 | 80.59 | 53.43 | 51.43 |
| | TIES | 29.20 | 11.76 | 84.85 | 81.24 | 53.07 | 52.02 |
| | DARE | 28.40 | 12.32 | 84.98 | 80.75 | 53.63 | 52.02 |
| | TSVM | 33.20 | 20.91 | 87.90 | 80.92 | 51.94 | 54.97 |
| | OrthoMerge | 33.60 | 18.17 | 89.25 | 81.98 | 52.66 | 55.13 |

*Table 8.* Performance of merging task-specific models fine-tuned from Qwen2.5-3B using LoRA and OFT across task-specific benchmarks.

## B.3. Performance Across Different Numbers of Tasks

Tables 9, 10, and 11 present the detailed results for all evaluated model merging methods when merging 2, 3, and 4 task-specific finetuned models, respectively. For each setting, we report the performance on multiple benchmarks, as well as the average score across tasks.

Across all experimental configurations, OrthoMerge consistently achieves the best or highly competitive average performance compared to alternative merging methods such as TA, TIES, DARE, and TSV-M. Notably, as the number of merged tasks increases, OrthoMerge maintains its robustness and effectiveness.

| Model | MATH500 | HUMANEVAL+ | Avg. |
|---|---|---|---|
| Llama-3.1-8B | 17.80 | 21.52 | 19.66 |
| Task-specific FT | 19.00 | 38.54 | 28.77 |
| TA | 23.60 | 43.65 | 33.63 |
| TIES | 23.20 | 43.04 | 33.12 |
| DARE | 22.80 | 40.55 | 31.68 |
| TSV-M | 21.80 | 42.32 | 32.06 |
| OrthoMerge | 21.80 | 47.25 | 34.53 |

*Table 9.* Comparison of different model merging methods across 2 tasks.

| Model | MATH500 | HUMANEVAL+ | ScienceQA | Avg. |
|---|---|---|---|---|
| Llama-3.1-8B | 17.80 | 21.52 | 6.12 | 15.15 |
| Task-specific FT | 19.00 | 38.54 | 26.75 | 28.09 |
| TA | 25.00 | 36.46 | 17.94 | 26.47 |
| TIES | 24.60 | 40.36 | 15.56 | 26.84 |
| DARE | 24.80 | 35.97 | 15.38 | 25.38 |
| TSV-M | 25.20 | 39.26 | 24.15 | 29.54 |
| OrthoMerge | 22.00 | 42.80 | 24.78 | 29.86 |

*Table 10.* Comparison of different model merging methods across 3 tasks.

| Model | MATH500 | HUMANEVAL+ | ScienceQA | CommonsenseQA | Avg. |
|---|---|---|---|---|---|
| Llama-3.1-8B | 17.80 | 21.52 | 6.12 | 70.60 | 29.01 |
| Task-specific FT | 19.00 | 38.54 | 26.75 | 82.47 | 41.69 |
| TA | 22.80 | 33.96 | 20.19 | 76.17 | 38.28 |
| TIES | 23.80 | 41.28 | 19.56 | 73.87 | 39.63 |
| DARE | 22.80 | 32.32 | 17.85 | 76.66 | 37.41 |
| TSV-M | 19.20 | 38.23 | 18.26 | 82.96 | 39.66 |
| OrthoMerge | 22.60 | 39.39 | 27.02 | 78.95 | 41.99 |

*Table 11.* Comparison of different model merging methods across 4 tasks.

## B.4. Visualization of Norm Distributions for Orthogonal-Residual Decoupling

To systematically analyze the norm distributions under the orthogonal-residual decoupling strategy applied to the models from Section 5.2, we computed and summarized the norms of the orthogonal component, residual component, and full task vector across different layers and module types. The visualizations include kernel density estimation to show the global distribution, scatter plots to illustrate the correlation between the orthogonal component and the full task vector, and bar and line plots to present the mean norms across module types and layers, respectively.

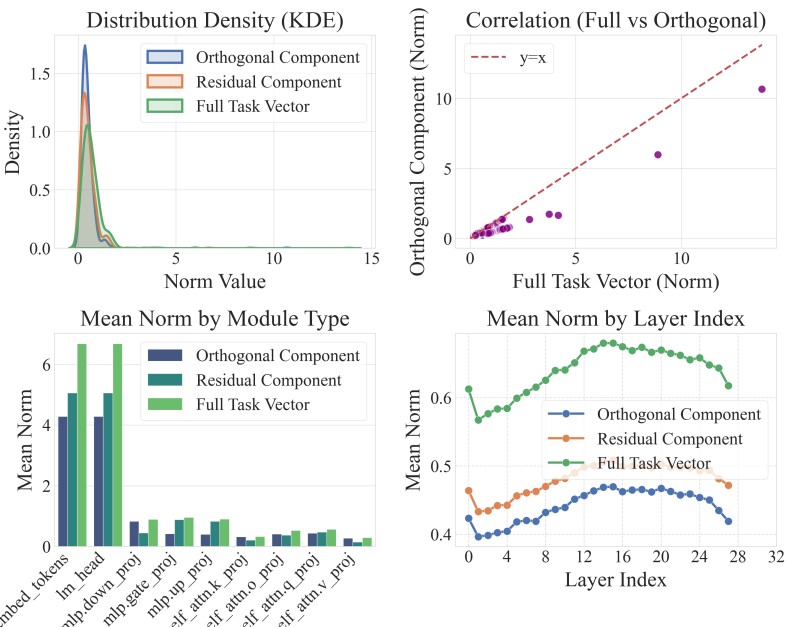

*Figure 6.* Distribution of the norms of orthogonal components of the models finetuned from Llama-3.2-3B from Section 5.2, obtained using the Global Decoupling strategy.

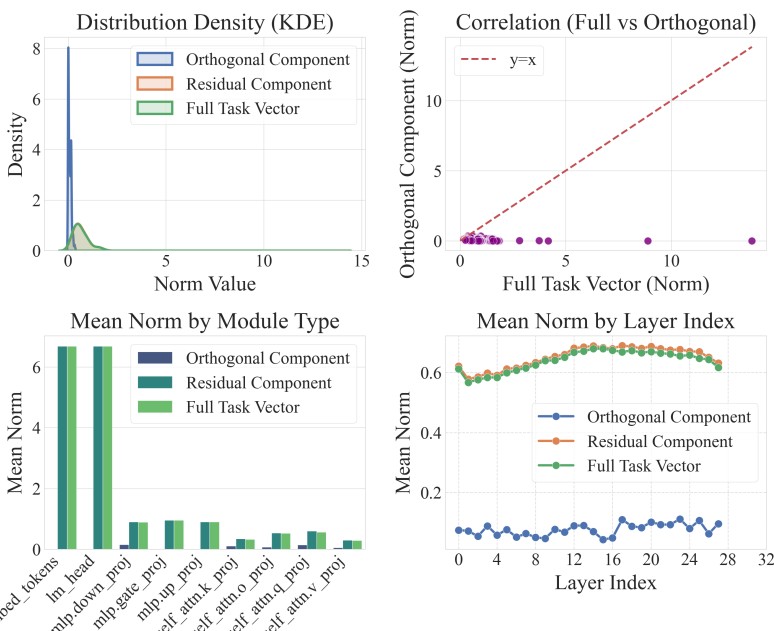

*Figure 7.* Distribution of the norms of orthogonal components of the models finetuned from Llama-3.2-3B from Section 5.2, obtained using the Conflict-Aware Decoupling strategy.

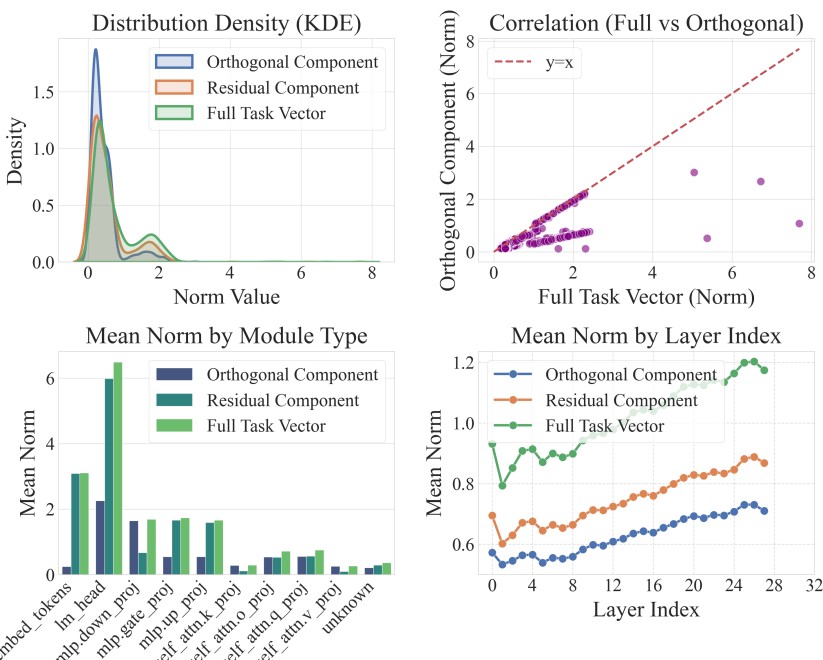

*Figure 8.* Distribution of the norms of orthogonal components of the models finetuned from Qwen2.5-VL-Instruct from Section 5.2, obtained using the Global Decoupling strategy.

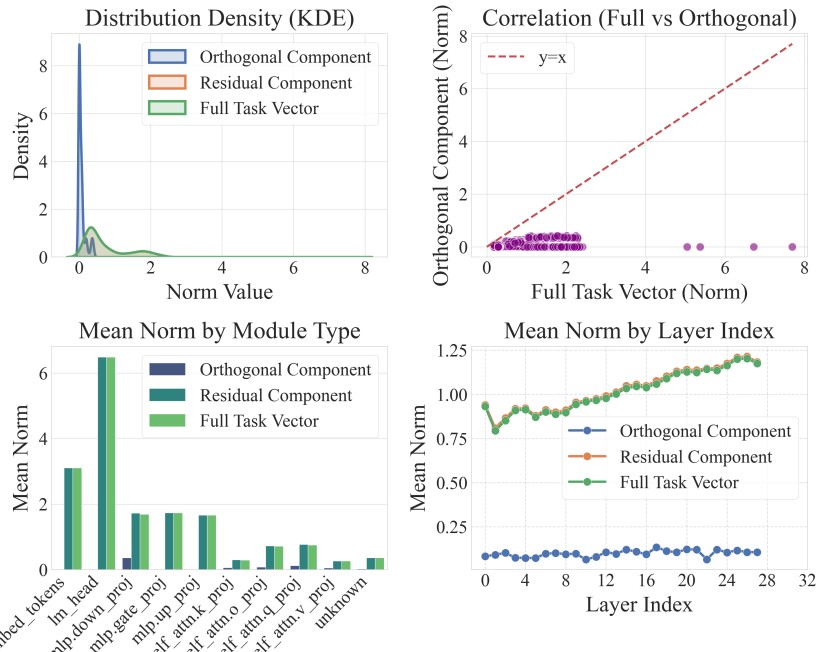

*Figure 9.* Distribution of the norms of orthogonal components of the models finetuned from Qwen2.5-VL-Instruct from Section 5.2, obtained using the Conflict-Aware Decoupling strategy.

