# OpenReview forum: "Orthogonal Model Merging"
_ICML.cc/2026/Conference — ICML 2026 regular_

### Official Review · Reviewer_tthp · 2026-03-07

**Soundness:** 3
**Presentation:** 3
**Significance:** 3
**Originality:** 3
**Overall Recommendation:** 4
**Confidence:** 4

**Summary:**

This paper studies a model merging method that preserves the orthogonality of model parameters. The method converts additive parameter updates into orthogonal multiplicative updates and designs an orthogonality-preserving strategy to perform merging. The proposed method improves the performance of the merged models under different fine-tuning strategies and base merging methods.

**Compliance With Llm Reviewing Policy:**

Affirmed.

**Final Justification:**

Despite the later discussion on the advantages of orthogonality, the paper lacks a clear motivation for introducing orthogonality. In the introduction, orthogonality is only briefly described as a “robust mechanism”, without theoretical justification or supporting citations. I sincerely encourage the authors to elaborate on the importance of orthogonality for model merging at the beginning of the paper, so as to help readers better understand the overall positioning and contributions of the work.

**Key Questions For Authors:**

It is unclear what the non-OFT methods refer to in the experiments. Moreover, it remains unclear whether the orthogonality-preserving strategy is effective for commonly used fine-tuning methods.

**Limitations:**

No, the authors have not adequately discussed the limitations of their work.

**Strengths And Weaknesses:**

### Strengths
- The paper is overall well-written and easy to follow.
- The paper presents a simple and effective method to improve model merging.

### Weaknesses
- The motivation of this paper is not sufficiently justified. Orthogonal fine-tuning is not a commonly adopted fine-tuning strategy, and the authors do not provide sufficient theoretical or empirical evidence to support the necessity of preserving orthogonality.
- The motivation behind the specific method design is also insufficient. The authors argue that the parameter merging process leads to a reduction in the magnitude of the skew-symmetric matrix $Q$, but they do not analyze the extent of this reduction or provide empirical evidence. Moreover, the impact of such magnitude reduction on model performance is not validated.
- The authors do not provide an analysis of the computational time or memory overhead of the proposed method.

---

> ### Author Rebuttal · Authors · 2026-03-31
>
> Thank you for your constructive comments. We appreciate your recognition that our paper is **"overall well-written and easy to follow"** and that it **"presents a simple and effective method to improve model merging."** We address each question below.
>
> > ***Weakness1:*** Insufficient justification for the necessity of preserving orthogonality.
>
> **A1:** Thank you for raising this point. We do not claim that preserving orthogonality is necessary. Rather, our motivation is that, compared with Euclidean fusion methods, orthogonality offers advantages for model weights, such as preserving norms and hyperspherical energy. Our experiments show the advantage of orthogonal fusion over Euclidean fusion. For non-OFT finetuned models, we adopt Orthogonal-Residual Decoupling. This decomposition of task vectors into the orthogonal component and the residual marries both advantages of orthogonal merging and Euclidean merging without the constraint of using orthogonal finetuning.
>
> > ***Weakness2:*** Lack of empirical analysis of magnitude reduction during merging and its impact on performance.
>
> **A2:** We provide concrete quantitative evidence below. For the experiments in Section 5.1, we measured the Frobenius norm of each task-specific adapter in Cayley space (averaged across layers):
>
> - **Qwen2.5-3B** — Individual adapter norms: [0.554, 1.008, 0.642, 1.638, 1.536]; mean: 1.076; norm after naive averaging: 0.561 (**48% reduction**)
> - **Llama-3.1-8B** — Individual adapter norms: [2.369, 2.571, 0.848, 1.276, 0.730]; mean: 1.559; norm after naive averaging: 0.833 (**47% reduction**)
>
> This substantial reduction confirms that naive averaging in the Lie algebra causes significant magnitude collapse. The ablation in Section 5.3 (Llama-3.1-8B, Simple Average Q vs. OrthoMerge) demonstrates its negative impact on performance. We additionally provide results for Qwen2.5-3B:
>
> | Model | MATH500 | HUMANEVAL+ | ScienceQA | CommonsenseQA | Social-IQA | Avg. |
> |---|:---:|:---:|:---:|:---:|:---:|:---:|
> | Qwen2.5-3B | 26.80 | 6.50 | 73.97 | 76.99 | 49.95 | 46.84 |
> | OrthoMerge | 33.60 | 18.17 | 89.25 | 81.98 | 52.66 | 55.13 |
> | −magnitude correction | 32.60 | 15.55 | 85.07 | 80.26 | 52.97 | 53.29 |
>
>
> > ***Weakness3:*** Missing analysis of the computational time and memory overhead.
>
> **A3:** For OFT models, OrthoMerge only requires simple operations in the Lie algebra space of the OFT adapters, and therefore has the smallest memory overhead among all methods. In terms of computational time, when merging five task-specific Qwen2.5-3B models, OrthoMerge takes 26.30 s, compared with 30.54 s for TA, 2 min 27 s for TIES, 2 min 33 s for DARE, and 18 min 15 s for TSV-M.
>
> For non-OFT models, for the Llama-3.2-3B experiments, solving the Procrustes problem per task-specific model takes 19 min 22 sec on a single GPU, with no additional memory overhead beyond baselines. Since model merging is a one-time offline procedure with no repeated computation at deployment, we consider this overhead acceptable in practice. To further improve efficiency, we implemented a fast variant using **Newton-Schulz iteration**, as in Muon [1], which reduces the time to **1 min 14 sec**. This fast variant even outperforms the SVD-based Procrustes approach (detailed results are provided in A1 of our response to Reviewer SKCw).
>
> [1] Keller Jordan, et al. "Muon: An optimizer for hidden layers in neural networks." Blog.
>
> > ***Question1 & Question2:*** Clarification on "non-OFT methods" and OrthoMerge's applicability to standard fine-tuning methods.
>
> **A4:** "Non-OFT methods" refers to standard fine-tuning approaches such as LoRA and full finetuning. Section 5.2 in the paper merges models obtained via full finetuning. To further validate our method on LoRA-finetuned models, we provide additional results on Qwen2.5-3B with LoRA adapters:
>
> | Model | MATH500 | HUMANEVAL+ | ScienceQA | CommonsenseQA | Social-IQA | Avg. |
> |---|:---:|:---:|:---:|:---:|:---:|:---:|
> | Qwen2.5-3B | 26.80 | 6.50 | 73.97 | 76.99 | 49.95 | 46.84 |
> | LoRA FT | 27.80 | 14.21 | 92.04 | 82.96 | 52.41 | 57.83 |
> | TA | 30.80 | 12.20 | 83.86 | 80.75 | 53.43 | 52.21 |
> | +OrthoMerge-C | 30.20 | 12.28 | 85.21 | 80.75 | 53.28 | 52.34 |
> | +OrthoMerge-G | 32.00 | 13.11 | 87.90 | 80.92 | 50.36 | 52.86 |
> | TIES | 31.60 | 21.03 | 83.59 | 81.24 | 53.89 | 54.27 |
> | +OrthoMerge-C | 31.40 | 21.69 | 83.45 | 81.75 | 53.57 | 54.37 |
> | +OrthoMerge-G | 31.40 | 20.19 | 87.86 | 80.84 | 51.20 | 54.30 |
> | TSV-M | 7.80 | 36.09 | 90.47 | 79.61 | 51.94 | 53.18 |
> | +OrthoMerge-C | 34.00 | 24.14 | 89.63 | 81.08 | 52.10 | 56.19 |
> | +OrthoMerge-G | 31.40 | 22.25 | 90.81 | 81.00 | 50.20 | 55.13 |
>
> LoRA FT represents finetuning each task individually using LoRA, serving as the performance ceiling for model merging. OrthoMerge consistently improves merging across all baselines, confirming that the benefit of orthogonal transformation is not limited to OFT models.
>
> Thank you once again for your time and valuable feedback!

---

> > ### Author Rebuttal · Reviewer_tthp · 2026-04-03
> >
> > I thank the authors for their rebuttal. My concerns have been resolved. I will increase my score accordingly.
> >
> > However, despite the later discussion on the advantages of orthogonality, the paper lacks a clear motivation for introducing orthogonality. In the introduction, orthogonality is only briefly described as a “robust mechanism”, without theoretical justification or supporting citations. I sincerely encourage the authors to elaborate on the importance of orthogonality for model merging at the beginning of the paper, so as to help readers better understand the overall positioning and contributions of the work.

---

> > > ### Author Response · Authors · 2026-04-04
> > >
> > > Thank you very much for your positive feedback and your willingness to increase your score. We will explain the importance of orthogonality for model merging at the beginning of the revised version.

---

### Official Review · Reviewer_JQrJ · 2026-03-11

**Soundness:** 3
**Presentation:** 3
**Significance:** 2
**Originality:** 2
**Overall Recommendation:** 5
**Confidence:** 4

**Summary:**

The paper proposes a merging strategy neamed Orthogonal Model Merging (OrthoMerging).
The authors proposed two variants of the method, one works with OFT adapters and the other works with LoRA adapters.
The first method proposes to derive the skew matrix $Q_i$ of the OFT, either directly via the OFT adapters R_i if they are available or from the finetuned weights, solving an Orthogonal Procustes problem. The $Q_i$ are then averaged, correcting the avergae with a costant $c$ that is the ratio between the sum of the frobenius norm of the $Q_i$ and the frobenius norm of the sum of the $Q_i$. Finally they reconstruct the $R_{merged}$ via Caley transform and obtain the merged model as $R_{merged}W_0$.
The second method, Orthogonal-Residual Decoupling, is a merging methods for merging non-OFT models. This method is based on the following ideas: cosnidering a target weight matrix, that can be either the  finetuned weights $W_i$ or a matrix obatin by the sum of the pretrained weights and the task matrix for task $i$ including only  directions conflicting with the average $\tau_{mean}$. As before solve an Orthogonal Procustes problem to obtained $R_i$ from the target matrix, and from this using the same procedure the first method obtain the $R_{merged}$. Finally they obtain the merged model as the sum of the component obtained form the $R_{merge}$  and a residual component performing the merging in the euclidian space.
The performances of the proposed method are compared  with the following baseline: TA, TIES, DARE, TSV-M.

**Compliance With Llm Reviewing Policy:**

Affirmed.

**Final Justification:**

The document is clearly written and well-structured; it analyzes a relevant problem and evaluates the proposed solution based on datasets, while the baseline models are appropriate. In the rebuttal phase the authors addressed my main concerns.

**Key Questions For Authors:**

Q1. How important is the scaling factor $c$ in practice, and would a similar magnitude correction help standard Euclidean task arithmetic as well?

Q2. In Figure 3, what exact loss is plotted (e.g., average over which tasks and datasets), and how is the joint loss defined?

**Limitations:**

yes

**Strengths And Weaknesses:**

### Strenghts
S1: The paper is clearly written and well structured

S2: The problem addressed is relevant and the solution found novel to the best of my knowledge

S3: The evaluation dataset and the baselines are reasonable

### Weaknesses

W1:  The way OrthoMerge is combined with baselines like TA, TIES, and TSV-M in Tables 3  is not clearly explained. Do you use these methods to merge the residuals together in standard Euclidean space?

W2: The theoretical justification for mixing manifold-based merging (orthogonal component) with Euclidean merging (residual component)  (eq 14) is mising.

W3: It would be interesting to add to Table 3 a comparison using only the orthogonal part (meaning the final model is just $W_{final} = R_{merged} W_0$, completely throwing away the residual $\rho_{merged}$).

**Minor**

mW1: It would be interesting to explain the differences in approach with respect to existing ‘task arithmetic in tangent space’ approach since they both share the core idea of mapping parameters into a tanget  space to safely perform linear arithmetic, even if the two tangent space are different.

mW2: Code it's not shared; the paper would benefit from a clear statement on code availability to ensure reproducibility

---

> ### Author Rebuttal · Authors · 2026-03-31
>
> Thank you for your constructive comments. We appreciate your recognition that our paper is **"clearly written and well structured,"** that **"the solution is novel,"** and that our **"evaluation dataset and baselines are reasonable."** We address each question below.
>
> > ***Weakness1:*** Whether residuals are merged in standard Euclidean space.
>
> **A1:** Yes. The residuals are merged in standard Euclidean space using these baseline methods.
>
> > ***Weakness2:*** Missing theoretical justification for mixing manifold-based merging (orthogonal component) with Euclidean merging (residual component) in Eq. 14.
>
> **A2:** The mixed formulation in Eq. 14 follows directly from the extraction process. Each expert weight is decomposed as **Wᵢ = Rᵢ · W₀ + ρᵢ**, where Rᵢ is the orthogonal component solved via Procrustes and ρᵢ is the residual. The merged model is therefore naturally expressed as **W_final = R_merged · W₀ + ρ_merged**, where R_merged is obtained by merging {Rᵢ} on the orthogonal manifold and ρ_merged by merging {ρᵢ} in Euclidean space.
>
> > ***Weakness3:*** Request for an ablation study in Table 3 using only the orthogonal component (discarding the residual).
>
> **A3:** We provide this ablation below. As expected, discarding the residual leads to a dramatic performance drop, confirming that the residual carries substantial task-specific information and that both components are essential. This is also because the original merged checkpoints are finetuned without any orthogonality constraints.
>
> | Model | Instruction | Math | Coding | Multilingual | Safety | Avg. |
> | :--- | :---: | :---: | :---: | :---: | :---: | :---: |
> | Llama-3.2-3B | 7.58 | 28.51 | 27.44 | 40.72 | 31.41 | 27.18 |
> | TA | 10.53 | 40.40 | 37.22 | 42.26 | 40.40 | 34.16 |
> | +OrthoMerge-C | 9.80 | 40.03 | 37.75 | 42.29 | 41.69 | 34.31 |
> | +OrthoMerge-C (no residual) | 7.65 | 28.80 | 27.10 | 40.73 | 31.97 | 27.25 |
> | +OrthoMerge-G | 11.09 | 43.90 | 37.96 | 42.06 | 42.41 | 35.48 |
> | +OrthoMerge-G (no residual) | 6.99 | 35.94 | 34.12 | 41.44 | 37.78 | 31.25 |
>
> > ***Question1:*** Importance of the scaling factor in practice, and whether a similar magnitude correction helps standard Euclidean task arithmetic.
>
> **A4:** The ablation in Section 5.3 (Llama-3.1-8B, Simple Average Q vs. OrthoMerge) demonstrates the importance of the scaling factor. We additionally provide results on Qwen2.5-3B:
>
> | Model | MATH500 | HUMANEVAL+ | ScienceQA | CommonsenseQA | Social-IQA | Avg. |
> |---|:---:|:---:|:---:|:---:|:---:|:---:|
> | Qwen2.5-3B | 26.80 | 6.50 | 73.97 | 76.99 | 49.95 | 46.84 |
> | OrthoMerge | 33.60 | 18.17 | 89.25 | 81.98 | 52.66 | 55.13 |
> | −magnitude correction | 32.60 | 15.55 | 85.07 | 80.26 | 52.97 | 53.29 |
>
> Moreover, applying magnitude correction to standard Euclidean TA also improves performance, but it still does not match OrthoMerge. This further suggests that our method benefits from using orthogonal transformations to better preserve weight norms and geometric structure.
>
> | Model | MATH500 | HUMANEVAL+ | ScienceQA | CommonsenseQA | Social-IQA | Avg. |
> |---|:---:|:---:|:---:|:---:|:---:|:---:|
> | Llama-3.1-8B | 17.80 | 21.52 | 6.12 | 70.60 | 48.11 | 32.83 |
> | Standard Euclidean TA | 21.00 | 34.20 | 29.90 | 80.18 | 54.15 | 43.89 |
> | +magnitude correction | 22.50 | 37.13 | 30.35 | 80.18 | 55.66 | 45.16 |
> | OrthoMerge | 24.60 | 38.90 | 32.01 | 80.59 | 55.17 | 46.25 |
>
>
> > ***Question2:*** Clarification on the exact loss plotted in Figure 3.
>
> **A5:** The loss in Figure 3 is the average loss over all the merged tasks, computed by randomly sampling an equal number of training examples from each task-specific dataset.
>
> > ***Minor Weakness1:*** Differences from existing "task arithmetic in tangent space" approaches.
>
> **A6:** While both methods leverage tangent spaces to safely perform linear arithmetic, they operate on fundamentally different manifolds and serve distinct purposes. The existing 'task arithmetic in the tangent space' approach linearizes the network function via the Neural Tangent Kernel (NTK), mapping the model into a functional tangent space to enhance weight disentanglement during finetuning. In contrast, our OrthoMerge operates strictly within the weight space geometry by mapping orthogonal weight matrices to their Lie algebra $\mathfrak{so}(d)$ (the tangent space of the orthogonal group manifold). Overall, our manuscript outlines the concept of explicitly preserving the intrinsic geometric constraints of the weights, such as hyperspherical energy, and spectral norm, whereas the prior work focuses on functional linearity to avoid task interference.
>
> > ***Minor Weakness2:*** Code availability for reproducibility.
>
> **A7:** As we are uncertain whether code release during the rebuttal period is permitted under the review policy, we promise the reviewer that the full codebase will be publicly released in our final version.
>
> Thank you once again for your time and valuable feedback!

---

> > ### Author Rebuttal · Reviewer_JQrJ · 2026-04-04
> >
> > Thank you for your response. My concerns have been addressed, I will adjust my score.

---

> > > ### Author Response · Authors · 2026-04-04
> > >
> > > We are glad that our response has addressed your concerns. Thank you for your willingness to adjust your score.

---

### Official Review · Reviewer_3y7M · 2026-03-13

**Soundness:** 3
**Presentation:** 3
**Significance:** 3
**Originality:** 3
**Overall Recommendation:** 5
**Confidence:** 3

**Summary:**

OrthoMerge does training-free merging on the orthogonal-group manifold instead of Euclidean weight arithmetic. For OFT, it merges orthogonal transforms via Lie-algebra space with a magnitude correction and maps back (Cayley). For non-OFT (LoRA / full finetuning), it proposes Orthogonal–Residual Decoupling: extract an “orthogonal component” via Procrustes, merge that on-manifold, and merge the residuals in Euclidean space. The paper reports experiments across multiple domains and includes ablations/analyses around the decoupling behavior.

**Compliance With Llm Reviewing Policy:**

Affirmed.

**Final Justification:**

I am raising my score to 5. The rebuttal effectively resolves my initial concerns. The wall-clock metrics and Newton-Schulz variant address the efficiency overhead. The mathematical proof that the orthogonal component naturally approaches the identity matrix (
I
) justifies the decoupling step in failure regimes, eliminating concerns about destructive rotations. Additionally, the new variance experiments confirm the method's stability. I recommend acceptance and urge the authors to incorporate these crucial updates into the final manuscript.

**Key Questions For Authors:**

1. **Practical efficiency:** Give wall-clock time, peak memory, and throughput/latency for OrthoMerge vs. TA/TIES/DARE/TSV-M. Include the non-OFT decoupling pipeline (per-block Procrustes/SVD). Parameter-count arguments are not enough.
2. **When does magnitude correction help/hurt?** You already have partial ablations (e.g., Table 4; C/G comparisons). What I’m missing is a systematic “failure regime” characterization: tasks / layers / adaptation strength where magnitude correction or C/G decoupling becomes harmful or unstable.
3. **Stability:** You vary task count in the appendix. Please add seed variance + task-subset sampling variance, and run the same stability checks specifically for the non-OFT decoupling pipeline.
4. **Decoupling validity:** Show concrete failure cases for Procrustes extraction (e.g., anisotropic / scaling-dominated / low-rank additive updates). I want to see direct evidence beyond norm/correlation plots.

**Limitations:**

No. The limitations/impact discussion is underdeveloped. It should state scope (when orthogonality assumptions are appropriate), decoupling failure modes, compute trade-offs, and potential misuse.

**Strengths And Weaknesses:**

The geometric angle is clear and nontrivial: if the adaptation lives near an orthogonal transform, Euclidean task-vector merging is the wrong geometry. The OFT part is clean, and the non-OFT extension is an interesting attempt to generalize the idea. Empirically, adding OrthoMerge variants tends to improve over standard Euclidean baselines.

The weak spot is the non-OFT story. The decoupling step is the paper’s main leap, and right now it’s not convincingly validated. Norm/correlation plots and Global vs. Conflict-Aware comparisons are not a substitute for showing when Procrustes extraction is actually meaningful—and when it fails. The magnitude-correction ablation suggests it matters, but there is still no clear boundary analysis (when it helps vs. hurts). Also, the “efficient integration” framing is under-supported: The paper does not provide a wall-clock, peak-memory, or throughput comparison with TA/TIES/DARE/TSV-M, despite requiring extra per-block Procrustes/SVD and Lie-algebra operations. Finally, the limitations/impact section is thin.

---

> ### Author Rebuttal · Authors · 2026-03-31
>
> Thank you for your constructive comments. We appreciate your recognition that our **"geometric angle is clear and nontrivial."** We are glad you found the OFT part **"clean,"** the non-OFT extension an **"interesting attempt to generalize the idea,"** and noted that empirically, **"adding OrthoMerge variants tends to improve over standard Euclidean baselines."** We address each question below.
>
> > ***Weakness1 & Question1:*** Lack of practical efficiency metrics.
>
> **A1:** For OFT models, OrthoMerge only requires simple operations in the Lie algebra space of the OFT adapters, and therefore has the smallest memory overhead among all methods. In terms of computational time, when merging five task-specific Qwen2.5-3B models, OrthoMerge takes 26.30 s, compared with 30.54 s for TA, 2 min 27 s for TIES, 2 min 33 s for DARE, and 18 min 15 s for TSV-M.
>
> For the Llama-3.2-3B experiments, orthogonal Procrustes takes 19 min 22 sec per task on one GPU. Since model merging is a one-time offline procedure with no repeated computation at deployment, we consider this overhead acceptable in practice. To further improve efficiency, we implemented a fast variant using **Newton-Schulz iteration**, as in Muon [1], which reduces the time to just **1 min 14 sec**. This fast variant even outperforms the SVD-based Procrustes approach (detailed results are provided in A1 of our response to Reviewer SKCw).
>
> All methods have the same inference throughput and latency, since the merge operation is not performed during inference. We will include these comparisons in the revised manuscript.
>
> [1] Keller Jordan, et al. "Muon: An optimizer for hidden layers in neural networks." Blog.
>
> > ***Weakness2 & Question4:*** Insufficient validation of the non-OFT decoupling step; need concrete failure cases for Procrustes extraction (e.g., anisotropic, scaling-dominated, or low-rank updates).
>
> **A2:** When a finetuned weight matrix undergoes purely anisotropic, scaling-dominated, or low-rank updates, the optimal solution to the orthogonal Procrustes problem naturally approaches the identity matrix $I$, because no rigid rotation can further reduce the residual error arising from non-rotational deformations. In this scenario, the orthogonal component contributes negligibly, and virtually all task-specific information is captured by the residual, with no destructive rotation is forced. This means OrthoMerge incurs no risk in such regimes.
>
> > ***Weakness3 & Question2:*** Lack of a systematic "failure regime" characterization detailing when magnitude correction or C/G decoupling becomes harmful or unstable.
>
> **A3:** Across all tasks evaluated, OrthoMerge consistently improves over the corresponding Euclidean baseline, and we did not observe cases in which magnitude correction or C/G decoupling led to degraded performance. We can offer the following observations: (1) For OrthoMerge-C, the orthogonal matrix norms in Cayley space are inherently small (as the conflicting component carries only a minority of task information), making magnitude correction contribute minimally; (2) when task-specific updates are identical, the averaging of skew-symmetric matrices experiences no destructive cancellation, so the magnitude correction factor becomes 1 (i.e., no adjustment is applied).
>
> > ***Question3:*** Stability checks (seed variance and task-subset sampling variance).
>
> **A4:** Since our merging process is deterministic, changing merging seeds does not affect results. For OFT models, we evaluated stability by varying training seeds prior to merging. For non-OFT models, altering training seeds is infeasible due to unreleased training details and datasets; however, we believe OrthoMerge's consistent superiority over baselines across diverse models already demonstrates its stability.
>
> **Seed Variance (OFT models, seeds 42/43/44) (Mean ± Std):** OrthoMerge achieves the highest and stable performance (**46.29** ± 0.12), outperforming TSV-M (44.95 ± 0.14), TIES (44.65 ± 0.17), TA (41.43 ± 0.10), and DARE (40.88 ± 0.14).
>
> **Task-Subset Variance:** **OFT (Llama3.1-8B):** On *{math, coding, science}*, OrthoMerge (**29.86**) > TSV-M (29.54) > TIES (26.84) > TA (26.47) > DARE (25.38). On *{commonsense, math, social}*, OrthoMerge (**52.31**) > TSV-M (51.89) > TIES (50.71) > DARE (50.06) > TA (48.43). **Non-OFT (Llama3.2-3B, Baseline -> +OrthoMerge-C / +OrthoMerge-G):** *{math, coding, instruction}:* TA (36.75 -> 37.28/38.90), TIES (40.80 -> 42.59/41.68), TSVM (42.82 -> 42.85/42.90). *{math, multilingual, instruction}:* TA (39.39 -> 41.42/42.02), TIES (47.16 -> 47.21/47.12), TSVM (46.16 -> 46.33/46.20).
>
> OrthoMerge demonstrates robust advantages across different seeds and task subsets.
>
>
> > ***Weakness4:*** The limitations and broader impact section is thin.
>
> **A5:** In the revised manuscript, we will expand the Limitations section.
>
> Thank you once again for your time and valuable feedback!

---

> > ### Author Rebuttal · Reviewer_3y7M · 2026-04-03
> >
> > Questions are properly answered. I would like to increase my score to 5.

---

> > > ### Author Response · Authors · 2026-04-04
> > >
> > > Thank you very much for your positive feedback and your willingness to increase your score.

---

### Official Review · Reviewer_SKCw · 2026-03-13

**Soundness:** 3
**Presentation:** 3
**Significance:** 2
**Originality:** 3
**Overall Recommendation:** 4
**Confidence:** 3

**Summary:**

This paper proposes OrthoMerge, a framework that performs model merging on the Riemannian manifold of the orthogonal group to preserve the intrinsic geometric properties of pretrained weights. For orthogonally fine-tuned (OFT) models, it aggregates updates within the Lie algebra using a magnitude-corrected scheme to prevent adaptation attenuation. To support non-OFT models, it introduces an Orthogonal-Residual Decoupling strategy that extracts structural rotations via Procrustes analysis for manifold merging, while handling the remaining linear residuals through standard Euclidean addition. Extensive experiments demonstrate that this geometry-aware approach effectively mitigates catastrophic forgetting and improves generalization across diverse language and multimodal tasks.

**Compliance With Llm Reviewing Policy:**

Affirmed.

**Final Justification:**

After considering the paper and rebuttal, the method is technically sound and offers a well-motivated geometric perspective on preserving feature structure during merging. The rebuttal addresses efficiency and design concerns, but questions remain regarding the necessity of strict orthogonality and the gap between theory and empirical behavior. Overall, the contribution is solid with some limitations. I increase my score to weak accept.

**Key Questions For Authors:**

* Since fine-tuning essentially operates within a very small subspace, would forcibly using SVD to extract a full-rank, large orthogonal matrix cause most useful information to remain in the residual?
* If manifold-based geometric fusion truly has universal advantages, why does it still rely on a purely empirical neuron-level selection mechanism? Does this suggest that when dealing with complex conflicts, pure geometric rotation alone cannot reliably guarantee the lower bound of task performance?

**Limitations:**

Yes.

**Strengths And Weaknesses:**

#### Strengths

* OrthoMerge proposes replacing simple translation (addition) with orthogonal transformations (rotations). Orthogonal transformations strictly preserve vector lengths and angles, thereby fundamentally protecting the feature structures that the pretrained model has painstakingly learned.
* Directly computing the geometric mean on a manifold is extremely difficult and time-consuming. The authors cleverly use the Cayley transform to convert a complex non-Euclidean geometric problem into a tractable linear algebra problem.
* In the Lie algebra space, a scalar factor is introduced to maintain the overall magnitude after merging, ensuring that the merged model does not “lose” fine-tuning strength due to task conflicts.

#### Weaknesses

* To extract orthogonal components from non-OFT models, the method must solve the orthogonal Procrustes problem, which is extremely computationally expensive and has a significant performance gap compared with matrix multiplication that is highly suitable for GPU parallel computation.
* The imposed “orthogonality” may not necessarily reflect the true nature of fine-tuning, as most useful information may remain in the residual components.
* Whether performing addition in Euclidean space or rotation on a Riemannian manifold, these methods essentially attempt to manipulate the weight space blindly in the hope that the function space will behave well. However, in highly nonlinear deep networks, preserving orthogonality in weight space does not mathematically guarantee optimal fusion of the output probability distributions (i.e., the model’s actual performance).

---

> ### Author Rebuttal · Authors · 2026-03-31
>
> Thank you for your constructive comments and for recognizing our method **"protecting feature structures with orthogonal transformations,"** the clever use of the **"Cayley transform to convert a complex non-Euclidean geometric problem into a tractable linear algebra problem,"** and the **"scalar factor to maintain the overall magnitude after merging."** We address each question below.
>
> > ***Weakness1:*** High computational cost and GPU inefficiency of solving the orthogonal Procrustes problem.
>
> **A1:** For Llama-3.2-3B, orthogonal Procrustes takes 19 min 22 sec per task on one GPU. Since model merging is a one-time offline procedure with no repeated computation at deployment, we consider this overhead acceptable in practice. To further improve efficiency, we implemented a fast variant using **Newton-Schulz iteration**, as in Muon [1], which reduces the time to just **1 min 14 sec**. This fast variant even outperforms the SVD-based Procrustes approach:
>
> | Model | Instruction | Math | Coding | Multilingual | Safety | Avg. |
> | :--- | :---: | :---: | :---: | :---: | :---: | :---: |
> | TA | 10.53 | 40.40 | 37.22 | 42.26 | 40.40 | 34.16 |
> | +OrthoMerge-C (Procrustes) | 9.80 | 40.03 | 37.75 | 42.29 | 41.69 | 34.31 |
> | +OrthoMerge-C (Newton-Schulz) | 10.72 | 39.88 | 37.63 | 42.23 | 41.12 | 34.32 |
> | +OrthoMerge-G (Procrustes) | 11.09 | 43.90 | 37.96 | 42.06 | 42.41 | 35.48 |
> | +OrthoMerge-G (Newton-Schulz) | 12.08 | 44.55 | 38.05 | 42.41 | 43.07 | 36.03 |
>
> [1] Keller Jordan, et al. "Muon: An optimizer for hidden layers in neural networks." Blog.
>
> > ***Weakness2 & Question1:*** The imposed "orthogonality" may not reflect the true nature of fine-tuning, potentially leaving useful information in the residual components.
>
> **A2:** Great question! For OFT-based merging, task information is forced into the orthogonal component. However, for non-OFT merging, we agree with the reviewer that the task information may not be fully captured by the orthogonal component. It is precisely the motivation behind our Orthogonal–Residual Decoupling design. We do not require the orthogonal component to capture all fine-tuning information. For OrthoMerge-C, the orthogonal matrix is designed to represent only the conflicting minority of task-specific changes; even so, orthogonal fusion consistently outperforms pure Euclidean fusion by mediating inter-task conflicts. The residual component captures the remainder through standard Euclidean merging, making the two designs complementary.
>
>
> > ***Weakness3:*** Preserving orthogonality in weight space does not mathematically guarantee optimal fusion in function space for highly nonlinear deep networks.
>
> **A3:** We agree with the reviewer on this limitation, but note that it is shared by all efficient and convenient data-free model merging methods (there is no guarantee for Euclidean merging neither). In deep learning, weight-space geometric constraints often serve as a powerful inductive bias, as guarantees do not hold in the highly non-linear function space. More importantly, our empirical results show that orthogonal components perform better than Euclidean components.
>
> A compelling example of this principle is the Muon optimizer. Muon explicitly orthogonalizes the momentum update matrices in the weight space before applying them to the network weights. Just like our merging approach, there is no strict mathematical proof that orthogonalizing updates in weight space guarantees an optimal trajectory in the function space. Yet, this constraint achieves SOTA training speeds for LLMs. In the same vein, our consistent gains over Euclidean baselines demonstrate the practical effectiveness of OrthoMerge.
>
> > ***Question2:*** Reliance on empirical neuron selection may suggest pure geometric rotation cannot reliably guarantee performance.
>
> **A4:** To assess the robustness of the conflict identification strategy, in addition to neuron-level conflict, we futher evaluated two other definitions: element-level conflict and TIES-defined conflict. We found that all yield consistent improvements over their respective baselines:
>
> | Model | Instruction | Math | Coding | Multilingual | Safety | Avg. |
> | :--- | :---: | :---: | :---: | :---: | :---: | :---: |
> | TA | 10.53 | 40.40 | 37.22 | 42.26 | 40.40 | 34.16 |
> | +OrthoMerge element-level conflicts | 11.28 | 40.18 | 37.25 | 42.29 | 40.43 | 34.29 |
> | TIES | 20.89 | 50.80 | 39.61 | 42.11 | 42.38 | 39.16 |
> | +OrthoMerge element-level conflicts | 23.84 | 55.12 | 39.83 | 42.21 | 46.05 | 41.41 |
> | +OrthoMerge TIES-defined conflicts | 23.48 | 55.04 | 39.61 | 42.22 | 43.33 | 40.74 |
> | TSV-M | 19.40 | 55.72 | 40.44 | 42.20 | 49.82 | 41.52 |
> | +OrthoMerge element-level conflicts | 24.40 | 53.07 | 40.75 | 42.60 | 49.06 | 41.98 |
>
> This demonstrates that the benefit of OrthoMerge is robust to the specific conflict definition, and stems from the orthogonal transformation rather than any particular selection heuristic.
>
> Thank you once again for your time and valuable feedback!

---

> > ### Author Rebuttal · Reviewer_SKCw · 2026-04-04
> >
> > Thank the authors for their response; some of my concerns have been addressed. If the core claim of this paper is that “strictly maintaining the geometric properties of weights on the orthogonal Riemannian manifold is key to improving fusion performance,” then the exact analytical solution (SVD) should, in principle, perform best. Why, then, does an approximate solution—obtained with only a few iterations and not fully orthogonal—achieve higher scores?

---

> > > ### Author Response · Authors · 2026-04-04
> > >
> > > Thank you very much for your continued constructive feedback and for raising this valuable question. We apologize that, due to the page limit, we did not analyze the experimental result in “A1” in the response. Here we provide the corresponding analysis and we will revise the paper accordingly.
> > >
> > > First of all, we would like to clarify that even in the original Procrustes procedure used in our paper, it is infeasible to obtain the exact analytical solution, because PyTorch’s SVD implementation is iterative and also computes an approximate solution based on Jacobi method (many Givens Rotation). While Jacobi method can be more precise than Newton-Schulz iteration, the additional precision is not necessarily helpful since the pretrained weights are stored in float16.
> > >
> > > We would also like to kindly point out that the experiment in “A1” corresponds to the Orthogonal-Residual Decoupling method for non-OFT models in our paper. The performance of the merged model obtained by Orthogonal-Residual Decoupling depends on the joint effect of both the orthogonal part and the residual part, rather than on the orthogonal part alone.
> > >
> > > While orthogonalization is the key, strict orthogonalization (especially more precision than the weights themselves) is not a necessary condition for achieving strong final performance in deep learning. For example, the recent Muon optimizer [1] uses orthogonalization as the key to process the gradients, but more precise orthogonalization may not necessarily be more helpful (more Newton-Schulz iterations can sometimes hurt the performance). Recent work [2] has also pointed this out. Possible reasons include the following:
> > > - Newton–Schulz produces a more conservative orthogonal component, which reduces over-rotation and leads to a better balance between structural alignment and task-specific adaptation.
> > > - The approximation error of Newton–Schulz may act as an implicit regularizer, preventing harmful noise from being overly absorbed into the orthogonal branch and thereby improving the robustness of the merged model.
> > >
> > > As a related example that showcases strictly better precision does not necessarily mean better generalization, it has been widely observed that mixed-precision training can often achieve better generalization than full-precision training.
> > >
> > > We will add the above analysis to the revised version, because we agree that this point is indeed very important.
> > >
> > > Again, thank you for your continued high-quality comments.
> > >
> > > [1] Jordon Keller et al. Muon: An optimizer for hidden layers in neural networks. 2024
> > >
> > > [2] Jack Zhang et al. Gram Newton-Schulz. 2026

---

### Decision · Program_Chairs · 2026-04-30

**Decision:**

Accept (regular)

**Comment:**

This paper proposes a new model merging method, termed Orthogonal Model Merging (OrthoMerge). The core idea is to preserve intrinsic geometric properties by pursuing orthogonality. In addition, a Newton–Schulz–based variant is introduced to accelerate the computation. All reviewers appreciate the general idea of the proposed method and acknowledge the accompanying theoretical discussion, resulting in overall positive evaluations.

At the same time, several concerns have been raised, which were further discussed during the AC–reviewer discussion. The most critical issue relates to the role of “orthogonality.” Based on both the theoretical analysis and the empirical results, orthogonality appears to be more as a heuristic rather than a well-established or rigorously validated principle. Therefore, the authors are strongly encouraged to moderate the corresponding claims to avoid potential overstatement.

Overall, my recommendation is accept, indicating that the paper makes a solid contribution, while some clarifications and refinements are expected in the final version.